# γδ T cells respond directly and selectively to the skin commensal yeast *Malassezia* for IL-17-dependent fungal control

Fiorella Ruchti[1,2], Meret Tuor[1,2], Liya Mathew[3], Neil E McCarthy[3], Salomé LeibundGut-Landmann [1,2]*

**1** Section of Immunology, Vetsuisse Faculty, University of Zürich, Zürich, Switzerland, **2** Institute of Experimental Immunology, University of Zürich, Zürich, Switzerland, **3** Centre for Immunobiology, Bart's and The London School of Medicine and Dentistry, The Blizard Institute, Queen Mary University of London, London, United Kingdom

* salome.leibundgut-landmann@uzh.ch

**Data Availability Statement:** All raw data linked to this study have been made publicly available on zenodo.org (doi: 10.5281/zenodo.10442783 and doi: 10.5281/zenodo.10449352).

## Abstract

Stable microbial colonization of the skin depends on tight control by the host immune system. The lipid-dependent yeast *Malassezia* typically colonizes skin as a harmless commensal and is subject to host type 17 immunosurveillance, but this fungus has also been associated with diverse skin pathologies in both humans and animals. Using a murine model of *Malassezia* exposure, we show that Vγ4+ dermal γδ T cells expand rapidly and are the major source of IL-17A mediating fungal control in colonized skin. A pool of memory-like *Malassezia*-responsive Vγ4+ T cells persisted in the skin, were enriched in draining lymph nodes even after fungal clearance, and were protective upon fungal re-exposure up to several weeks later. Induction of γδT17 immunity depended on IL-23 and IL-1 family cytokine signalling, whereas Toll-like and C-type lectin receptors were dispensable. Furthermore, Vγ4+ T cells from *Malassezia*-exposed hosts were able to respond directly and selectively to *Malassezia*-derived ligands, independently of antigen-presenting host cells. The fungal moieties detected were shared across diverse species of the *Malassezia* genus, but not conserved in other Basidiomycota or Ascomycota. These data provide novel mechanistic insight into the induction and maintenance of type 17 immunosurveillance of skin commensal colonization that has significant implications for cutaneous health.

## Author summary

*Malassezia* is the most abundant fungus living on our skin and is usually harmless, but this microbe has also been shown to play a role in pathological conditions such as eczema and dermatitis. Here, we investigated how a population of Vγ4+ γδ T cells protects mouse skin against fungal overgrowth. While generally considered part of the innate immune system, we found that γδ T cells were maintained long after *Malassezia* was cleared from the skin of experimentally-infected animals. These cells displayed memory-like features and were highly efficient at fighting the fungus after a secondary challenge. We observed

**Funding:** This work was supported by the Swiss National Science Foundation (grant # 310030_189255 to SLL), a Career Development Award from The Medical Research Council (MR/R008302/1 to NEM), and project grant funding from Bart's and The London Charity (MGU0465 to NEM). The funders had no role in study design, data collection and analysis, decision to publish, or preparation of the manuscript.

**Competing interests:** I have read the journal's policy and the authors of this manuscript have the following competing interests: NEM has received consultancy fees and funding for research from ImCheck Therapeutics SAS, as well as funding for research from TC BioPharm.

that classic fungal pattern recognition receptors were not involved, and that antigen-presenting cells were not required to generate *Malassezia*-protective γδ T cells. Finally, we confirmed that murine γδ T cells can directly recognise *Malassezia* and that the *Malassezia*-derived structure that triggers these responses was not conserved among other fungi. Our results highlight for the first time an important role for γδ T cells in preventing uncontrolled growth of the most abundant skin fungus. These findings have implications for several *Malassezia*-associated pathologies including eczema, dermatitis, and potentially even cancer.

## Introduction

γδ T cells are important mediators of tissue homeostasis and wound healing in the skin, which acts as a critical barrier against damage and infection. Skin-resident γδ T cells are also in constant crosstalk with the microbiota and exert critical functions in host defense [1–3]. While γδ T cells arise from the same precursors as other lymphocytes, unlike αβ T cells they acquire effector functions in the thymus before seeding epithelial tissues pre-natally and during early life [4]. Generally, γδ T cells are considered innate lymphocytes because of their lack of MHC restriction. In mice, they are developmentally pre-programmed and respond rapidly to cytokines and microbes [4,5]. More recently, adaptive-like functions such as memory formation and extra-thymic differentiation have been attributed to γδ T cells [6]. The most abundant γδ T cell populations in murine skin are monoclonal Vγ5+ dendritic epidermal T cells (DETCs) in the outmost layers, and oligoclonal subsets expressing Vγ4 or Vγ6 T cell receptor (TCR) chains in the dermis, whereas peripheral lymphoid organs contain a more polyclonal pool of Vγ4+ cells [7]. Murine dermal γδ T cells predominantly express IL-17 cytokine, which plays an important role in barrier repair and antimicrobial defense of epithelial tissues [7]. Consequently, dermal γδ T cells have been implicated in host protection against various cutaneous bacteria and fungi, as shown in mouse models of infection with *Mycobacterium bovis* [8], *Staphylococcus aureus* [9,10], *Candida albicans* [11], *Candida auris* [12], and *Trichophyton mentagrophytes* [11,13]. In addition to mediating protective immunity, IL-17-producing γδ T cells have been implicated in inflammatory skin disorders such as psoriasis and contact dermatitis [2,6]. Still, many aspects of γδ T cell activation, tissue-residency versus recruitment, and host protection versus pathogenicity remain unknown.

*Malassezia* is by far the most abundant skin-resident fungal genus in humans and other warm-blooded animals [14]. While *Malassezia* is classified within the phylum Basidiomycota and most closely related to the plant pathogen *Ustilago maydis*, this yeast also shares genetic and functional similarities with other skin-colonizing fungi such as *Candida albicans*, indicating host-specific adaptation [15]. *Malassezia* is a lipid-dependent yeast preferentially colonizing sebaceous skin sites [16] and interacts with the cutaneous environment by secreting lipases, phospholipases, proteases, and bioactive indoles. Some of these secreted factors have pro-inflammatory features, such as *M. furfur*-derived indoles, which have been implicated in the development of seborrheic dermatitis (SD) [17], and proteases that have been linked to barrier disruption and impaired wound healing [18]. As such, *Malassezia* is not only a commensal, but can also display pathogenic characteristics depending on context. In addition to SD, *Malassezia* has been associated with dandruff, pityriasis versicolor, and atopic dermatitis (AD) [19]. A number of allergens have been described in *M. sympodialis*, with around half of adult AD patients displaying sensitization to *M. sympodialis* and other *Malassezia* species [20,21]. In veterinary dermatology, *M. pachydermatis* is frequently associated with canine

atopic eczema and is a common agent of ear infections [22]. However, the factors that determine the relative role of *Malassezia* in health versus disease remain largely unknown.

Antifungal immunity is critical to maintain stable colonization and establish commensalism. In healthy individuals, the homeostatic response against *Malassezia* is dominated by IL-17-producing T cells [23,24]. Likewise, in an experimental model of *Malassezia* skin colonisation, type 17 immunity is critical for preventing fungal overgrowth [24]. Here we identified dermal Vγ4+ γδ T cells as the predominant cellular source of IL-17 in *Malassezia*-colonized murine skin. We investigated the cellular and molecular mechanisms underlying this γδT17 response and observed that IL-17 was induced in a *Malassezia*-specific and selective manner. While initial activation depended on IL-1 and IL-23, host cytokines were redundant for re-activation of *Malassezia*-induced Vγ4+ γδ T cells. Rather, soluble *Malassezia*-derived factors were sufficient to elicit IL-17 release in previously activated Vγ4+ γδ T cells. These soluble fungal stimuli were conserved across the *Malassezia* genus but not shared with other fungi.

## Results

### Cutaneous immunity against *Malassezia* depends on IL-17A and IL-17F cytokines

To dissect the cellular and molecular mediators of *Malassezia*-induced type 17 immunity, we employed an experimental model of fungal skin colonization in C57BL/6 mice [24]. We first investigated the dynamics of IL-17 induction following fungal exposure and then assessed long-term dependence on this cytokine for cutaneous fungal control. As early as 2 days after epicutaneously associating the unperturbed mouse ear with *M. pachydermatis*, *Il17a* and *Il17f* transcripts were strongly induced in the skin tissue, as reported previously [24]. Transcript levels further increased on days 4 and 7 (Fig 1A). The IL-17 response was accompanied by mild inflammation manifesting in increased ear thickness (Fig 1B and 1C) and infiltration of neutrophils into the skin as observed by histology (Figs 1C **and** S1A) quantified by flow cytometry (Figs 1D **and** S1B). Both parameters were strongly reduced in IL-17A/F-deficient mice, highlighting the IL-17-dependence of this *Malassezia*-induced response (Fig 1B–1D). Skin fungal loads were elevated in *Il17af*$^{-/-}$ mice in comparison to their wild type (WT) counterparts as early as day 2 after colonization, emphasizing the rapid kinetics and critical role of IL-17 in mediating fungal control. In WT mice the fungus was cleared to undetectable levels by day 12, whereas in *Il17af*$^{-/-}$ mice the fungal burden persisted for at least 60 days (Fig 1E). The colonization load stabilized at intermediate levels after a drop during the second week, which may be due to neutrophils and/or other effector mechanisms that act at least partially independently of IL-17. Overall, these data confirm the essential and long-lasting role of IL-17A/F in skin immunity to *M. pachydermatis*.

### Vγ4+ γδ T cells are the primary IL-17A producers in *Malassezia*-associated skin

To identify the source of IL-17A in the *M. pachydermatis*-colonized skin, we first made use of IL-17A-eYFP fate reporter mice (*ll17a*$^{Cre}$ *R26R*$^{eYFP}$ mice [25]). Flow cytometry-based analysis of eYFP expression coincided with IL-17A protein expression, thus confirming specificity of the reporter (S2A Fig). While the eYFP signal was very low in vehicle-treated control mice, eYFP expression was strongly induced in response to *M. pachydermatis* skin association (S2B Fig). The eYFP signal was restricted to CD90+ lymphocytes and most abundant in γδ T cells, especially within the Vγ4+ dermal subset (S2A, S2C and S2D Fig). On day 7, TCRγδ-negative T cells also contributed to the overall eYFP+ population, although Vγ4+ γδ T cells remained

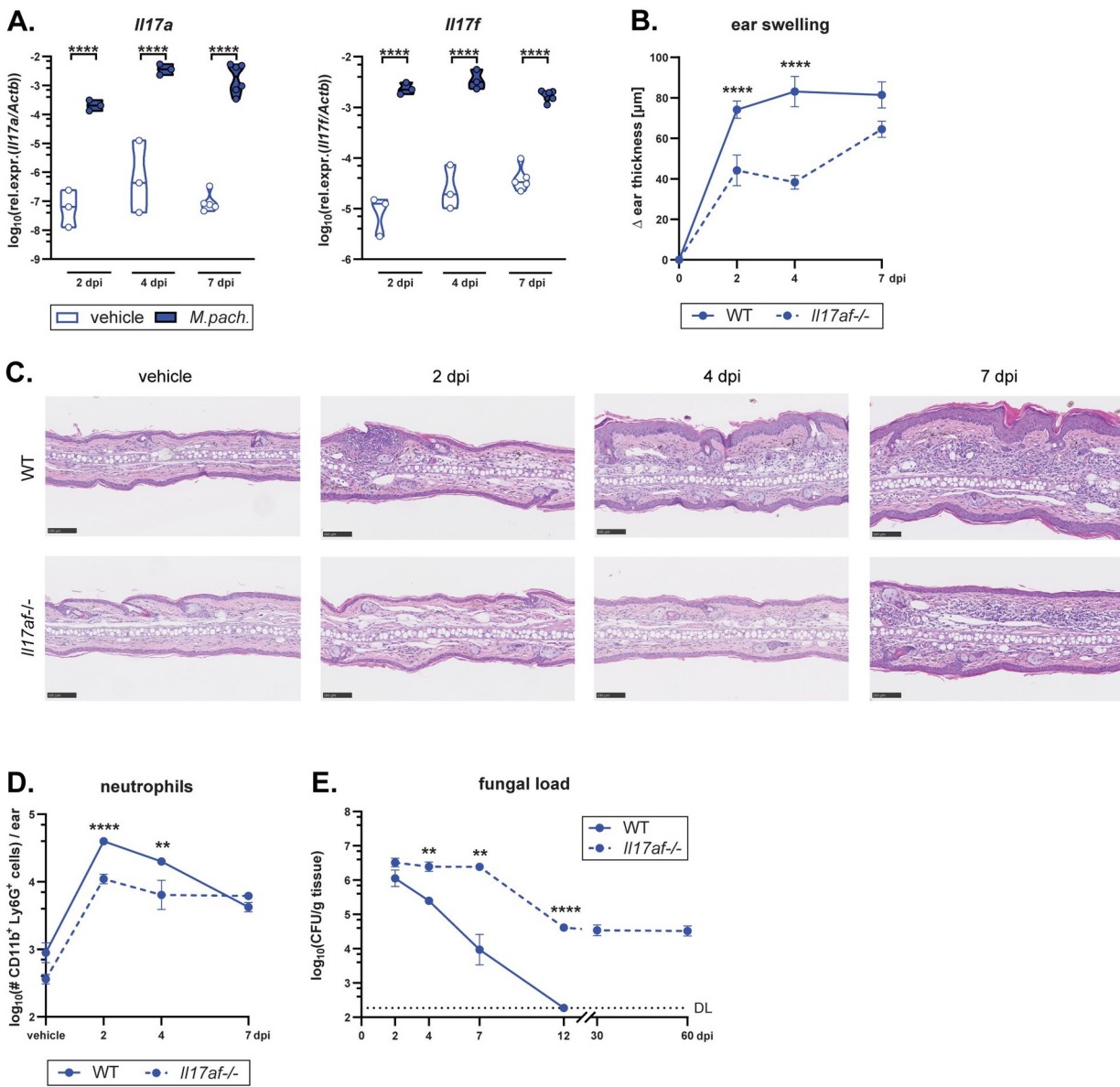

**Fig 1. Cutaneous immunity against *Malassezia* depends on IL-17A and IL-17F cytokines.** The ear skin of wild type (WT) and *Il17af^{-/-}* mice was associated with *M. pachydermatis* or treated with olive oil (vehicle control) and analysed at the indicated time points (days post infection, dpi). **A.** *Il17a* and *Il17f* transcript levels in the *M. pachydermatis*-associated and control ear skin of WT mice. **B.** Increase of ear thickness in WT and *Il17af^{-/-}* mice. **C.** Hematoxylin and eosin-stained ear tissue sections. **D.** Skin neutrophil numbers. **E.** Skin fungal load (CFU). DL, detection limit. Data in A-D are from two independent experiments with 2–4 mice per group. Data in E are compiled from five independent experiments with at least 3 mice per group and time point (except at 60 dpi, which is 2 mice per group). In A, the median of each group is indicated. In B, D and E, the mean +/- SEM of each group is indicated. Statistical significance was determined using two-way ANOVA. **p<0.01, ****p<0.000**1. See also** S1 Fig.

the largest IL-17A-expressing subset in *M. pachydermatis*-associated skin (S2D Fig). These findings were recapitulated by direct staining for IL-17A protein in ear skin cells after re-stimulation of single cell suspensions with PMA and ionomycin. Overall IL-17A-producing cells, and IL-17A-producing Vγ4+ γδ T cells in particular, gradually increased in numbers over the course of a week (Fig 2A–2C). The overall size of the Vγ4+ γδ T cell population, including both

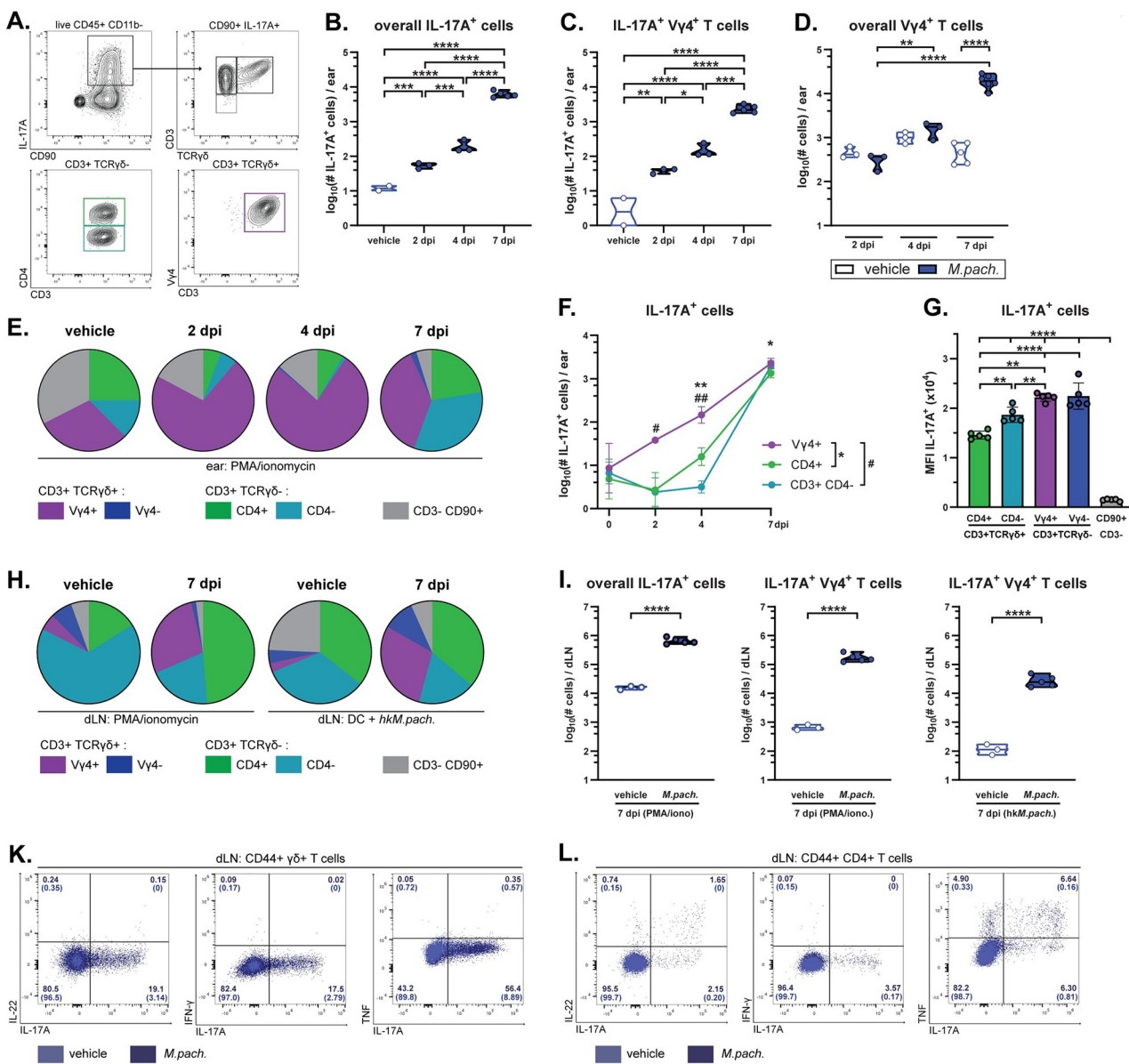

**Fig 2. Vγ4⁺ γδ T cells are the main IL-17A producers in *Malassezia*-associated skin.** The ear skin of WT mice was associated with *M. pachydermatis* or treated with olive oil (vehicle control) and IL-17 production in skin and draining lymph nodes was analysed by flow cytometry at the indicated time points (dpi) or on day 7 (if not specified otherwise). **A.** Gating strategy for identifying the IL-17⁺ cellular subsets among viable CD45⁺ CD11b⁻ single cells in the skin. **B.-C.** Quantification of IL-17A⁺ cells among overall skin CD90⁺ cells (B) or among Vγ4⁺ TCRγδ⁺ CD90⁺ T cells (C) after ex vivo re-stimulation with PMA and ionomycin. **D.** Quantification of non-restimulated Vγ4⁺ γδ T cells in the ear skin. **E.-G.** Proportion (E), total numbers (F), and IL-17A median fluorescence intensity (G) of the indicated IL-17A producing cell populations in the ear skin after ex vivo re-stimulation with PMA and ionomycin. **H.-I.** Proportion (H) and total numbers (I) of the indicated IL-17A-producing CD44⁺ cell populations in the ear-draining lymph nodes (dLN) after re-stimulation with PMA and ionomycin or with heat-killed *M. pachydermatis* (hk*M.pach.*)-pulsed DCs, as indicated. Data in A-I are from one representative of two independent experiments with 3–5 mice per group. In B, C, D, G and I, each symbol represents one animal. In B, C, D and I, the median of each group is shown. In F and G, the mean +/- SD of each group is shown. **K.-L.** IL-22, IFN-γ or TNF co-production with IL-17 by Vγ4⁺ (K) and CD4⁺ T cells (L) in the dLN after re-stimulation with hk*M.pach.* pulsed DCs in *M. pachydermatis*-associated (dark blue) or vehicle-treated control mice (light blue). Four or two concatenated samples from associated or control mice, respectively, are shown with numbers in quadrants indicating percentages of CD44⁺ TCRγδ⁺ (K) or CD4⁺ (L). Data for IL-22 and IFN-γ are from day 7 in one of two representative experiments while TNF is from one experiment on day 5. Statistical significance was determined using one-way ANOVA (B, C, F, G), two-way ANOVA (D) or unpaired Student's *t* test (I). *p<0.05, **p<0.01, ***p<0.001, ****p<0.0001. **See also** S2 Fig.

IL-17$^+$ and IL-17$^-$ subsets, remained relatively stable until day 4 after fungal association before sharply increasing relative to the vehicle control group (Fig 2D). Vγ4$^+$ γδ T cells represented the largest IL-17$^+$ T cell subset in the skin at all time points analyzed (Fig 2E and 2F). On a per cell basis, Vγ4$^+$ γδ T cells also expressed higher levels of IL-17A than any other T cell subset or CD90$^+$ CD3$^-$ innate lymphoid cells, which made a negligible contribution (Fig 2G). *Malassezia*-responsive γδ T cell expansion was comparable to that achieved on stimulation with PMA and ionomycin (S2E Fig). Likewise, in the ear-draining lymph nodes (dLN) Vγ4$^+$ γδ T cells represented a large portion of the overall IL-17A-producing pool, whether re-stimulated with PMA and ionomycin, or in an fungus-specific manner using dendritic cells (DCs) pulsed with heat-killed *Malassezia* (Fig 2H and 2I). This strong induction of IL-17A-producing Vγ4$^+$ γδ T cells was not restricted to *M. pachydermatis* skin exposure, since this response was also prominent after challenge with *M. sympodialis* or *M. furfur* (S2F–S2I Fig). While Th17 cells readily co-produced IL-22 and TNF (Fig 2L), γδ T cells could not be observed to express IL-22, IFN-γ or TNF in response to *M. pachydermatis* (Fig 2K). In line with previous analyses of γδT17 cells at barrier sites [4], *M. pachydermatis*-elicited γδT17 cells were CD103$^+$, CCR6$^+$ and CD27$^-$ (S2K and S2L Fig). In summary, we identified Vγ4$^+$ γδ T cells as the main IL-17A-producing cell subset in the *Malassezia*-associated skin and as a prominent population in the dLN.

## γδ T cells are key mediators of antifungal immunity to *Malassezia*

After identifying Vγ4$^+$ γδ T cells as a key source of IL-17A in the *M. pachydermatis*-associated skin, we assessed the consequences of γδ T cell deficiency on infection dynamics and the overall antifungal response. *Tcrd*$^{-/-}$ mice [26] manifested strongly impaired IL-17 responses to the fungus, and from day 4 to day 7 no other cellular subset was able to fully compensate for the absence of γδ T cells (Fig 3A and 3B). Likewise, IL-17-driven inflammation, as reflected by increased ear thickness, was markedly decreased in *Tcrd*$^{-/-}$ mice compared to their WT counterparts (Fig 3C). This became further apparent by analyzing hematoxylin and eosin-stained histology sections from the ear skin of *Malassezia*-associated *Tcrd*$^{-/-}$ mice, which exhibited lower epidermal thickness and reduced tissue inflammation (Fig 3D), and by quantifying neutrophil numbers in the *M. pachydermatis*-colonized skin (Fig 3E). Finally, mice lacking γδ T cells displayed a marked inability to control fungal loads by day 14, a time point at which αβ T cells had long begun to produce IL-17 (Figs 2E, 2F and 3F). Still, colonization levels declined over time and by day 21 only low levels of *Malassezia* were detected, while by day 30 the fungus was fully cleared presumably by γδ T cell-independent compensatory sources of IL-17 (Fig 3F). In conclusion, γδ T cells are essential for IL-17-production and fungal control early after *Malassezia* skin association and for several weeks.

## A pool of γδ T cells in lymph nodes supports long-term protection against *Malassezia*

While γδ T cells share several characteristics with prototypical innate immune cells, they can exhibit features of immunological memory [27]. Given the long-lasting protective effects of γδ T cells observed in *Malassezia*-colonized skin, we wondered whether these cells may exhibit enhanced activity upon a secondary fungal encounter. To assess this, we used WT mice that had been exposed or not to the fungus 41 days earlier, then challenged with *M. pachydermatis* for a further 3 days before analysing γδ T cell responses (Fig 4A). As a control we included a group of mice that were associated with the fungus only once, at the initial time point of infection. Skin inflammation was not significantly impacted by exposure history since ear swelling, total skin leukocytes, and neutrophil counts were comparably high in both groups (Figs 4B and S3A–S3C). A single exposure 44 days prior to analysis had no impact on neutrophil

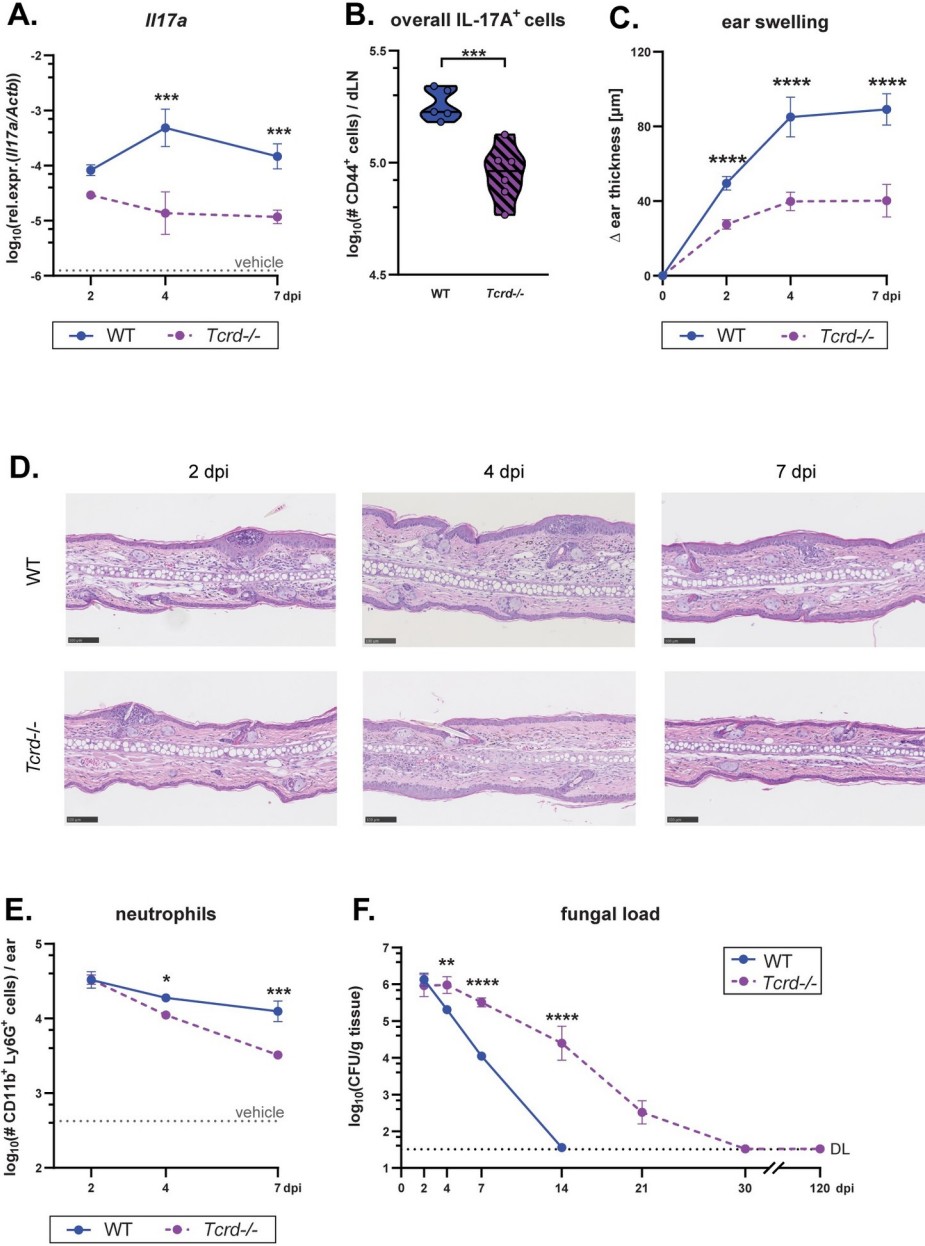

**Fig 3. γδ T cells are key mediators of antifungal immunity to *Malassezia*.** The ear skin of WT and *Tcrd*[-/-] mice was associated with *M. pachydermatis* or treated with olive oil (vehicle control) and analysed at the indicated time points (dpi). **A.** IL-17A transcript levels in the ear skin. **B.** Quantification of IL-17A[+] cells among overall CD44[+] CD90[+] draining lymph node (dLN) cells at 7 dpi after re-stimulation with heat-killed *M. pachydermatis* (hk*M.pach*.)-pulsed DCs. **C.** Increase in ear thickness. **D.** Hematoxylin and eosin-stained ear tissue sections. **E.** Skin neutrophil numbers. **F.** Skin fungal load (CFU). Data in A, C and E are pooled from, and data in B and D are from one representative of two independent experiments with 3–6 mice per group. Data in F are compiled from five independent experiments with 2–6 mice per group resulting in 4–9 mice per time point except for the 30 and 120 dpi time points which are 2 and 3 mice per group, respectively. The mean +/- SEM of each group is shown in A, C, E and F. The median of each group is shown in B. DL, detection limit. Statistical significance was determined using two-way ANOVA (A, C, E and F) or unpaired Student's *t* test (B). *p<0.05, **p<0.01, ***p<0.001, ****p<0.0001.

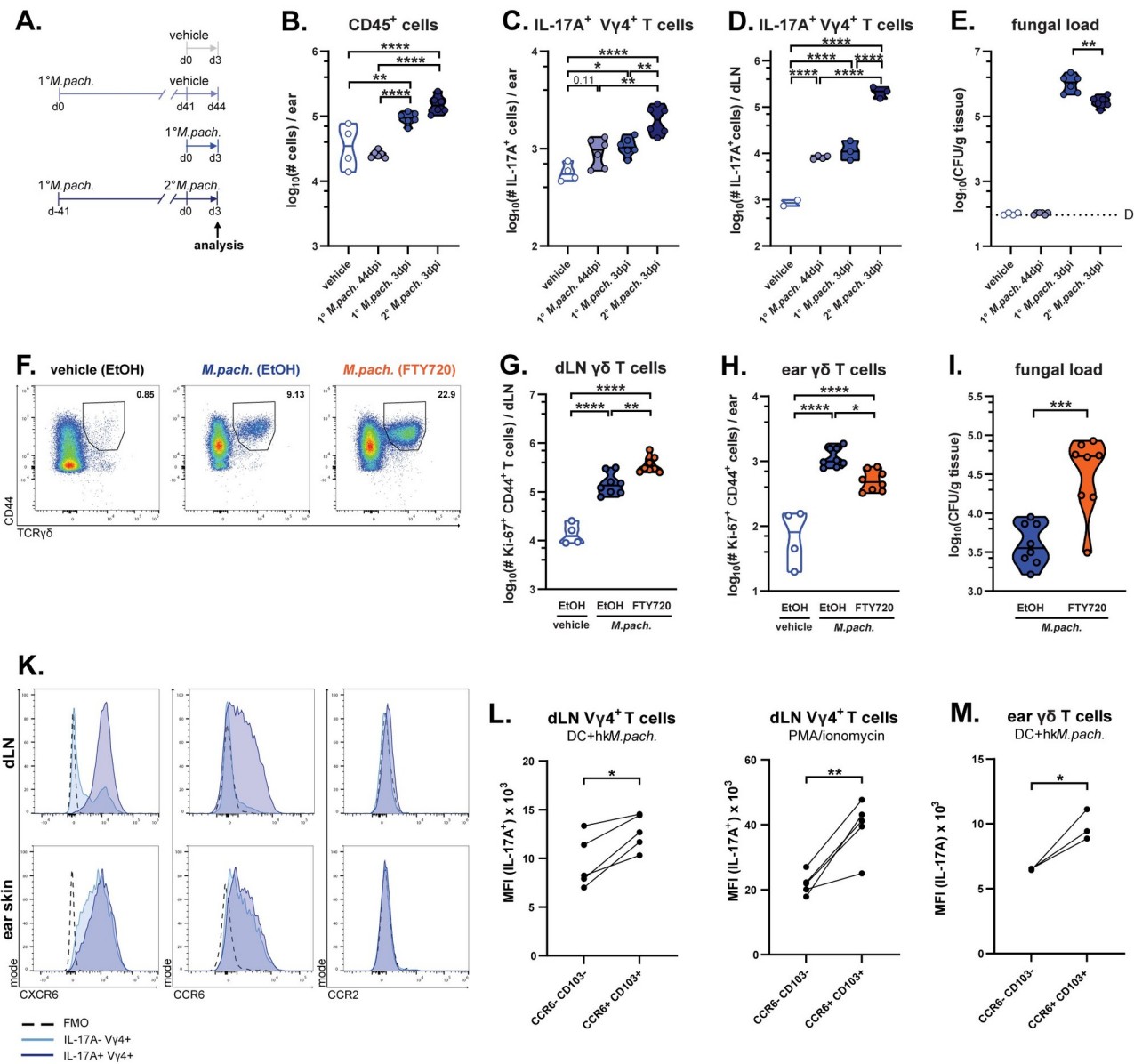

**Fig 4. A pool of γδ T cells in lymph nodes supports long-term protection against *Malassezia*. A.-E.** WT mice were associated with *M. pachydermatis* (*M.pach.*) once or twice or treated with olive oil (vehicle control) according to the scheme in (A). Total numbers of overall CD45$^+$ cells (B) and IL-17$^+$ Vγ4$^+$ γδ T cells in ear skin (C) and IL-17A$^+$ Vγ4$^+$ γδ T cells in dLN after re-stimulation with heat-killed *M. pachydermatis*-pulsed DCs (D). Skin fungal load (E). Data in B, C and E are compiled from two independent experiments, and data in D are from one representative of two independent experiments with 2–4 mice per group. Each symbol represents one mouse. The median of each group is indicated. **F.-I.** WT mice were associated with *M. pachydermatis* for 7 days and treated with FTY720 or EtOH (solvent control) in drinking water from day -1 until day 7 of colonisation. Representative flow cytometry plots showing CD44$^+$ TCRγδ$^+$ cells among viable CD45$^+$ CD90$^+$ Ki-67$^-$ dLN cells (F). Total numbers of Ki-67$^+$ γδ T cells in dLN (G) and ear skin (H) and fungal loads (CFU) in the skin (I). Data in G-I are pooled from two independent experiments with 2–4 mice per group. Each symbol represents one mouse. The median of each group is indicated. **K.-M.** Chemokine receptor expression by IL-17$^+$ and IL-17$^-$ Vγ4$^+$ γδ T cells in dLN and skin of *M. pachydermatis*-associated WT mice after re-stimulation with heat-killed *M. pachydermatis*-pulsed DCs or PMA and ionomycin, as indicated. Histograms showing three concatenated samples from one representative of of two (dLN) or three (ear) independent experiments (K) including a fluorescence minus one (FMO) control per experiment. Median fluorescence intensity of IL-17A staining in the CCR6$^-$ CD103$^-$ and CCR6$^+$ CD103$^+$ subsets of Vγ4$^+$ γδ T cells in dLN (L) and ear skin (M) Data are from one representative of three (L) or two (M) independent experiments. Connected data points are from the same mouse. Statistical significance was determined using one-way ANOVA (B-E, G, H) or unpaired (I) or paired (L, M) Student's *t* test. *p<0.05, **p<0.01, ***p<0.001, ****p<0.0001. **See also** S3 Fig.

infiltration of the skin compared to a group that had never previously encountered the fungus (vehicle control) (Figs 4B **and** S3C). When assessing the lymphocyte compartment, we observed that IL-17-producing Vγ4+ γδ and CD4+ T cells were increased in the skin 3 days after fungal association. However, only skin Vγ4+ γδT17 cells and not Th17 cells displayed pronounced expansion upon fungal re-exposure in comparison to primary encounter (Figs 4C **and** S3D). IL-17 production in the LN was also markedly increased after fungal re-exposure, by both, *Malassezia*-responsive γδ T cells and CD4+ T cells (Figs 4D, S3D and S3E). The enhanced response of skin γδ T cells in mice that had been previously exposed to *M. pachydermatis* coincided with accelerated fungal control, which was not evident in animals exposed for the first time 3 days prior (Fig 4E). Intriguingly, in mice that had been associated with fungus and then rested for 44 days without further challenge, *Malassezia*-responsive Vγ4+ γδ T cell numbers in dLN remained 10-fold higher than in vehicle-treated controls, whereas CD4+ T cells returned to baseline levels in the same animals (Figs 4D **and** S3E). Therefore, it appears that a significant pool of *Malassezia*-responsive Vγ4+ γδ T cells is maintained in the dLN, and to a limited degree also in the skin, even after fungal clearance.

Having observed a persistent pool of *Malassezia*-responsive Vγ4+ γδ T cells in the dLN of previously exposed animals, and rapid accumulation of these cells in the dLN after fungal exposure, we next assessed whether fungus-responsive dermal Vγ4+ γδ T cells depend on expansion in and trafficking from the dLN. To test this hypothesis, we blocked egress of lymphocytes from the lymph nodes using fingolimod (FTY720). Indeed, proliferating γδ T cells were decreased in the ear skin of FTY720-treated mice relative to ethanol (EtOH)-treated controls 7 days after *Malassezia* association, while concomitantly accumulating in the dLN (Fig 4F–4H). Consistent with these γδ T cell dynamics (Fig 2D) no such effect was observed on day 4 (S3F and S3G Fig) and ear swelling remained unaffected (S3H Fig). Consequently, FTY720-mediated blockade of trafficking resulted in impaired fungal control by day 7 (Fig 4I), but not by day 4 (S3I Fig). Hence, migration of IL-17 producing lymphocytes, including the predominant γδ T cell fraction and possibly other T cells, is required for efficient antifungal control. In support of γδ T cells trafficking from the dLN to the skin, IL-17 production by dLN Vγ4+ γδ T cells in response to *M. pachydermatis* correlated with expression of the skin-homing chemokine receptors CXCR6 and CCR6, as well as the tissue residency marker CD103 (Fig 4K). Moreover, the highest IL-17 levels were observed in the CCR6+ CD103+ subset of Vγ4+ γδ T cells in both LN and skin in response to polyclonal and/or antigen-specific restimulation (Fig 4L and 4M). In contrast, CCR2 was not expressed despite previously being linked with γδ T cell recruitment during inflammation and autoimmunity [28]. In the *Malassezia*-colonized ear skin, all Vγ4+ γδ T cells expressed high levels of CXCR6 and CCR6 (Fig 4K). Together, these results support the notion that Vγ4+ γδ T cells persisting in the dLN after fungal exposure can expand and traffic to the skin for prolonged local fungal control.

## γδT17 cell responses to *Malassezia* do not depend on CD11c+ cells or T cell receptor signalling

Next, we aimed at identifying the signals that drive skin γδ T cell responses to *Malassezia*. Several different mechanisms have been proposed to activate dermal γδ T cell in both microbial and non-microbial contexts [29]. Having observed a prominent increase in Vγ4+ γδ T cell in the dLN, we speculated that antigen-presenting cells may play a role in γδ T cell activation. To test this, we depleted CD11c+ cells prior to *Malassezia* skin association by administering diphtheria toxin (DT) to bone-marrow chimeric mice expressing the DT receptor under control of the CD11c promoter (CD11c-DTR [30]) in the radio-sensitive compartment. While *Malassezia*-responsive Th17 cells were strongly reduced in the transgene positive (TG+) group

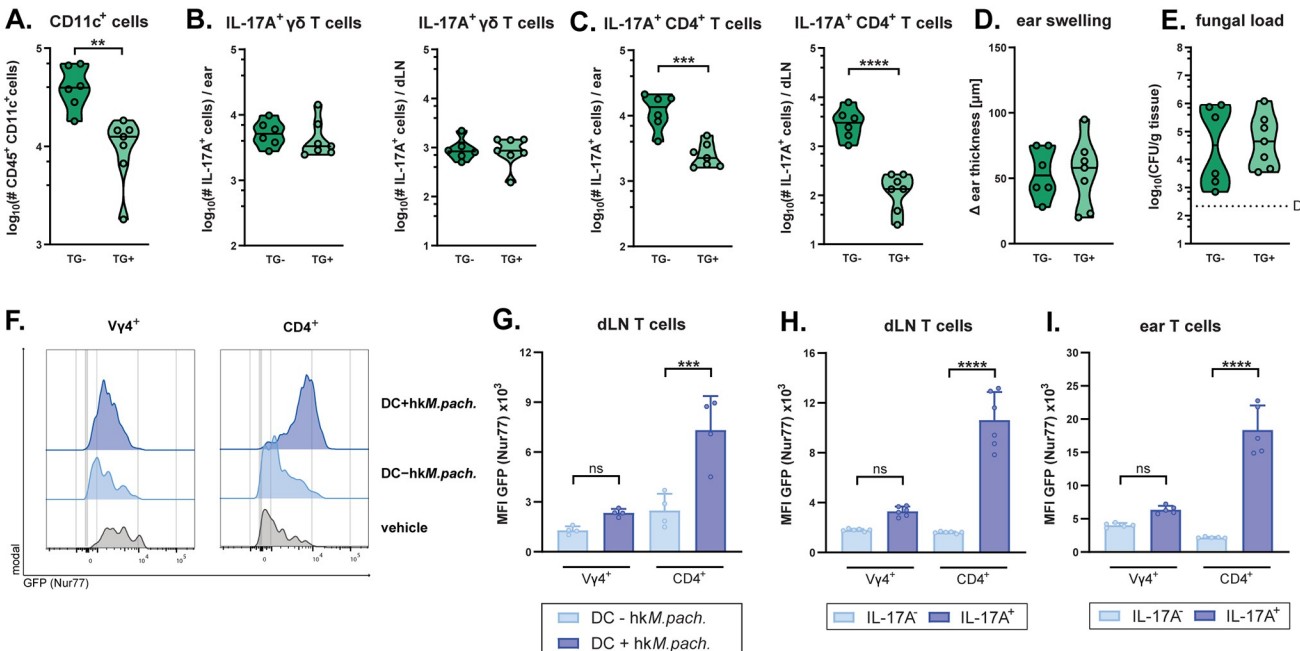

**Fig 5. γδT17 cell responses to *Malassezia* do not depend on CD11c+ cells or T cell receptor signalling. A.-E.** Irradiation chimeras reconstituted with CD11c-DTR transgene positive (TG+) or transgene negative (TG-) bone marrow as a control were treated with diphtheria toxin prior to association with *M. pachydermatis* for 7 days. Total numbers of CD45+ CD11c+ cells in the dLN (A). Total numbers of IL-17A+ γδ T cells (B) and IL-17A+ CD4+ T cells (C) in the ear skin (left) and dLN (right) after re-stimulation with heat-killed *M. pachdermatis*-pulsed DCs (in case of dLN T cells) or with PMA and ionomycin (in case of skin T cells), respectively. Increase in ear thickness (D). Skin fungal loads (CFU) at 7 dpi (E). Data are pooled from two independent experiments with three mice per group. Each symbol represents one mouse. The median of each group is indicated. DL, detection limit. **F.-I.** Nur77GFP mice were associated with *M. pachydermatis* or treated with olive oil (vehicle control) for 7 days and GFP expression by γδ T cells and CD4+ T cells in the dLN or ear was assessed after restimulation with DCs that were or were not pulsed with heat-killed *M. pachydermatis* (DC +/- hk*M.pach.*). Histograms are from three (vehicle) or six (*M.pachydermatis*-associated) concatenated samples (E). GFP median fluorescence intensity (MFI) in CD44+ Vγ4+ γδ T cells and CD4+ T cells (**G**). GFP MFI of IL-17+ and IL-17- cells after restimulation with *M. pachdermatis*-pulsed DCs in dLN (**H**) and ear skin (**I**). Data are from one representative of two independent experiments with four to six mice per group. Bars in **G-I** are the mean+SD of each group. Statistical significance was determined using unpaired Student's *t* test (A-**E**) or one-way ANOVA (**G-I**) **p<0.01, ***p<0.001, ****p<0.0001.

compared to the transgene negative (TG-) group upon DT treatment, CD11c+ cell depletion did not impair IL-17-production by γδ T cells in the ear skin or dLN (Fig 5A–5C) and ear swelling was unchanged (Fig 5D). In line with our observation that γδ T cells confer protection from *Malassezia* overgrowth at 7 dpi, fungal control also remained intact after depletion of CD11c+ cells (Fig 5E). To obtain further evidence that the *Malassezia* responsiveness of Vγ4+ γδ T cells was independent of antigen presentation and TCR signalling, we next made use of Nur77 reporter (Nur77GFP) mice [31]. Nur 77 is an immediate early gene downstream of the TCR [31]. Flow cytometry-based quantification of Nur77-driven GFP expression in T cells from these mice allows determining the degree of antigen receptor engagement. Seven days after fungal exposure, dLN lymphocytes were co-cultured with DCs that had been pulsed or not with heat-killed *M. pachydermatis*, which induced a high degree of TCR signalling in CD4+ T cells but not in γδ T cells (Fig 5F and 5G). A similar pattern of TCR engagement was observed when comparing IL-17A+ versus IL-17A- dLN cells or ear tissue-derived T cells after re-stimulation with *M. pachydermatis*-pulsed DCs (Fig 5H and 5I). Taken together, these results indicate that Vγ4+ γδ T do not rely on antigen presenting cells and exhibit very limited TCR signalling during activation and IL-17 responses to *Malassezia*.

## γδ T cells are activated independently of C-type lectin and Toll-like receptor signalling in response to *Malassezia*

γδ T cells express diverse pattern recognition receptors (PRRs) and have been shown to respond to direct PRR stimulation [5]. As for other fungi, the cell wall of *Malassezia* can be recognized by C-type lectin receptors (CLR), especially Dectin-2 and Mincle [32–34]. Therefore, we assessed which receptors mediate the γδ T cell response to *Malassezia*. However, mice lacking Dectin-1 [35] or Dectin-2 [36] or chimeras with a Mincle-defective [37] hematopoietic compartment displayed no reduction in γδ T cell population size, or in the number of IL-17-producing γδ T cells in skin and dLN after *M. pachydermatis* skin association (S4A–S4I Fig). Even mice lacking Card9, the central adapter protein required for CLR signalling [38], did not display any defect in γδ T cell-derived IL-17 responses to *M. pachydermatis* (Fig 6A and 6B) or in inflammatory response as quantified by overall ear swelling (S4K–S4N Fig). These data indicate that CLRs are not required for the antifungal γδ T cell response in this context, regardless of potential compensatory effects between individual receptors that may occur in single CLR-deficient mice. Consequently, we did not observe an impairment of fungal control in the skin of Card9- and CLR-deficient mice (Figs 6C, S4C, S4F and S4I), with the only exception being Dectin-2 knockout mice (S4F Fig). This finding suggests that Dectin-2 may mediate IL-17-independent antifungal effector functions, possibly within the myeloid compartment. Indeed, bone marrow-derived dendritic cells from dectin-2-, but not dectin-1-, deficient mice were previously reported to display impaired expression of multiple cytokines and chemokines, and production of reactive oxygen species in response to *Malassezia* [32]. Next, we tested the involvement of Toll-like receptor (TLR) signalling in γδ T cell responses to *M.*

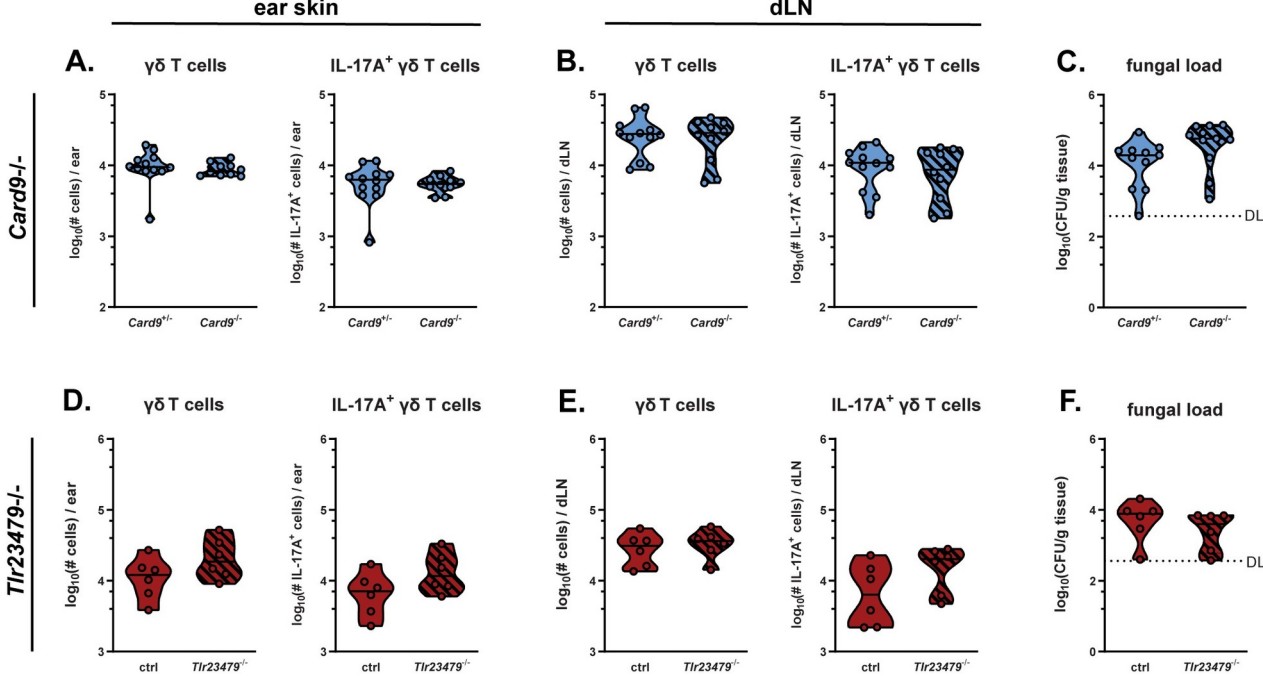

**Fig 6. γδ T cells are activated independently of C-type lectin and Toll-like receptor signalling in response to *Malassezia*. A.-I.** The ear skin of *Card9*[-/-] and *Card9*[+/-] littermate control (A-C) and *Tlr23479*[-/-] and control (D-F) mice was associated with *M. pachydermatis*. After 7 days, γδ T cells in skin and dLN were quantified (A, D) and analyzed for IL-17 production (B, E). The fungal load (CFU) was assessed in the skin (C, F). Data are pooled from three (A-C) or two (D-F) independent experiments with 3–5 mice per group. Each symbol represents one mouse. The median of each group is indicated. DL, detection limit. Statistics were calculated using unpaired Student's *t* test. **See also** S4 Fig.

*pachydermatis*. We made use of mice lacking five major TLRs (*Tlr23479*[-/-] mice [39]), among which TLR2 was previously shown to respond to *Malassezia* spp. in keratinocytes and mast cells [40, 41] and TLR3 can be engaged by mycovirus-containing *Malassezia* [42,43]. Following *M. pachydermatis* skin association, ear swelling, overall γδ T cell numbers and γδ T cell-derived IL-17 production in skin and dLN of *Tlr23479*[-/-] mice was conserved at WT levels, and likewise skin fungal load was unchanged compared with controls (Figs 6D–6F **and** S4O). In summary, neither CARD9-dependent CLR nor TLR signalling are required to elicit an effective γδ T cell response to *Malassezia*.

## The antifungal γδ T cell response depends on IL-23 and IL-1 family cytokine signalling

In addition to PRRs, γδT17 cells express diverse cytokine receptors among which IL-1R and IL-23R drive population activation and expansion [5,44], and support *de novo* thymus-independent generation of these cells [45,46]. MyD88 plays a critical role in TLR pathways but also mediates IL-1 family cytokine signalling [47], and mice deficient in this central adaptor protein displayed strongly impaired γδ T c

ell responses to *Malassezia* exposure (Fig 7A and 7B). When assessed at 7 dpi, total and IL-17A-producing γδ T cell numbers were reduced in skin and dLN of fungus-colonized MyD88-deficient mice when compared to heterozygous littermate controls (Fig 7A and 7B). Consistent with the host-protective role of IL-17-producing γδ T cells (Fig 3F), skin fungal load was increased in *Myd88*[-/-] vs. *Myd88*[+/-] mice [48] (Fig 7C). Having already excluded TLR-mediated effects (Fig 6D–6F), we attributed the MyD88-dependence of the γδ T cell response against *Malassezia* to an involvement of IL-1 family cytokines.

Similar results were obtained with mice deficient in IL-23R signalling. When assessed at 7 dpi, both total and IL-17-producing γδ T cell numbers in the skin and dLN of *Il23r*[-/-] mice resembled those in naïve mice (Fig 7D and 7E). Consequently, fungal loads were strongly increased in *Il23r*[-/-] relative to *Il23r*[+/-] control mice (Fig 7F). Similar results were also obtained in *Il23a*-deficient mice (S5A and S5B Fig). Furthermore, *Il23r*[-/-] animals displayed less tissue inflammation as quantified by ear swelling compared to littermate controls (S5C Fig), phenocopying our earlier findings in *Tcrd*[-/-] mice (Fig 3C). In contrast, there was no significant difference in ear swelling in the absence of functional MyD88 signalling (S5D Fig). In summary, these results show that IL-23 is indispensable for the induction of γδT17 cells and effective control of *Malassezia* colonisation, whereas IL-1 contributes only partially to γδ T cell-mediated immunity.

Neutrophils have been identified as a major source of IL-23 in the *Malassezia*-associated skin [24]. Antibody-mediated depletion of Gr-1[+] cells including neutrophils significantly impaired fungal control (Fig 7G and 7H) resulting in decreased tissue expression levels of *Il23a* and *Il1b* after 3 days of *Malassezia* skin association (Fig 7I–7K). Of note, neutrophil recruitment to the *Malassezia*-associated skin at 7 dpi was not impacted by the lack of IL-23R signalling. This finding implies that *Malassezia*-induced skin inflammation is mediated by IL-23 effects on γδ T cells rather than neutrophils (S5C and S5E Fig). In conclusion, IL-23 and IL-1 family cytokines are important for efficient γδT17 immunity against *Malassezia*, while neutrophils may further support γδ T cell function by enhancing early provision of IL-1 and IL-23 cytokines.

## γδ T cells respond directly and specifically to *Malassezia*-derived structures

Having identified key roles for IL-1 and IL-23 in initiating γδT17 cell immunity against *Malassezia*, we next investigated whether Vγ4[+] γδ T cells from *M. pachydermatis*-exposed mice can

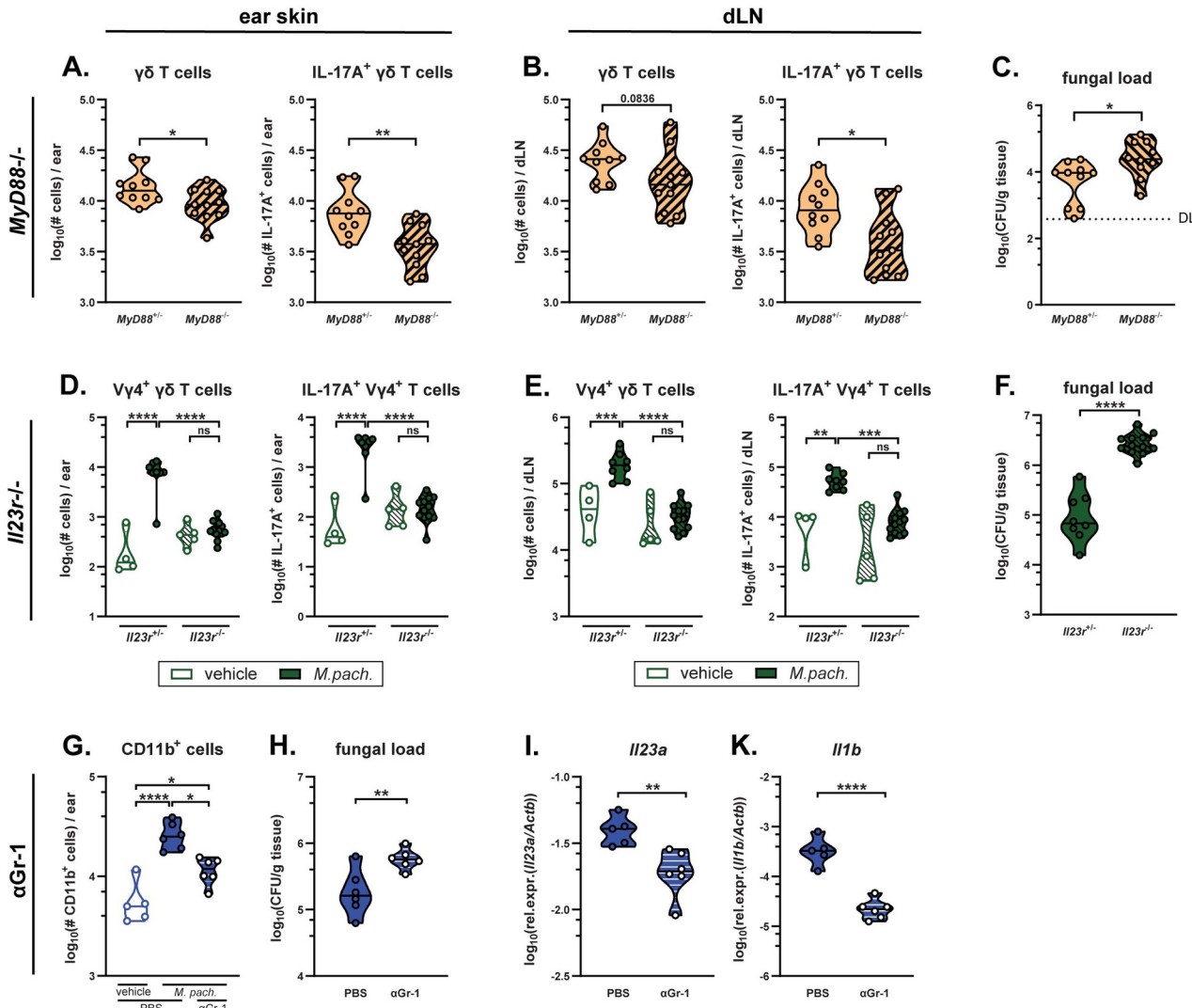

**Fig 7. The antifungal γδ T cell response depends on IL-23 and IL-1 family cytokine signalling. A.-F.** The ear skin of *MyD88*[+/-] and *MyD88*[-/-] littermate control (A-C) and *Il23r*[-/-] and *Il23r*[+/-] littermate control (D-F) mice was associated with *M. pachydermatis*. Vehicle treated groups were included for *Il23r*[-/-] and *Il23r*[+/-] mice. After 7 days, skin and dLN γδ T cells were quantified (A, D) and analyzed for IL-17 production (B, E), and the skin fungal load (CFU) was assessed (C, F). **G.-K.** Mice were treated daily with anti-Gr-1 depletion antibody (αGr-1) or control (PBS) starting one day prior to association with *M. pachydermatis*. Skin CD11b[+] cell numbers (G), fungal load (H), transcript levels of *Il23a* (I) and *Il1b* (K) were assessed at 3 dpi. Data are pooled from three (A-F) or two (G-K) independent experiments with 2–5 mice per group. Each symbol represents one mouse. The median of each group is indicated. DL, detection limit. Statistics were calculated using unpaired Student's *t* test (A-C, F, H-L), one-way ANOVA (G) or two-way ANOVA (D-E). *p<0.05, **p<0.01, ***p<0.001, ****p<0.0001. **See also** S5 Fig.

respond to these cytokines directly. For this, we FACS-purified γδ T cells (S6A and S6B Fig) from the dLN of *M. pachydermatis*-associated mice, of which almost all Vγ4[+] γδ T cells produced IL-17A after PMA restimulation. When restimulating with recombinant cytokines, IL-17 secretion was elicited in response to high concentrations of IL-1α and IL-1β, whereas IL-23 alone was not sufficient (Fig 8A and 8B). The IL-1-triggered IL-17 response was comparable to γδ T cell reactivation by DCs that had been pulsed with heat-killed *M. pachydermatis* (Fig 8A and 8B), as routinely tested in previous experiments. We therefore hypothesized that *Malassezia*-pulsed DCs release cytokines that act on Vγ4[+] γδ T cells rather than providing TCR

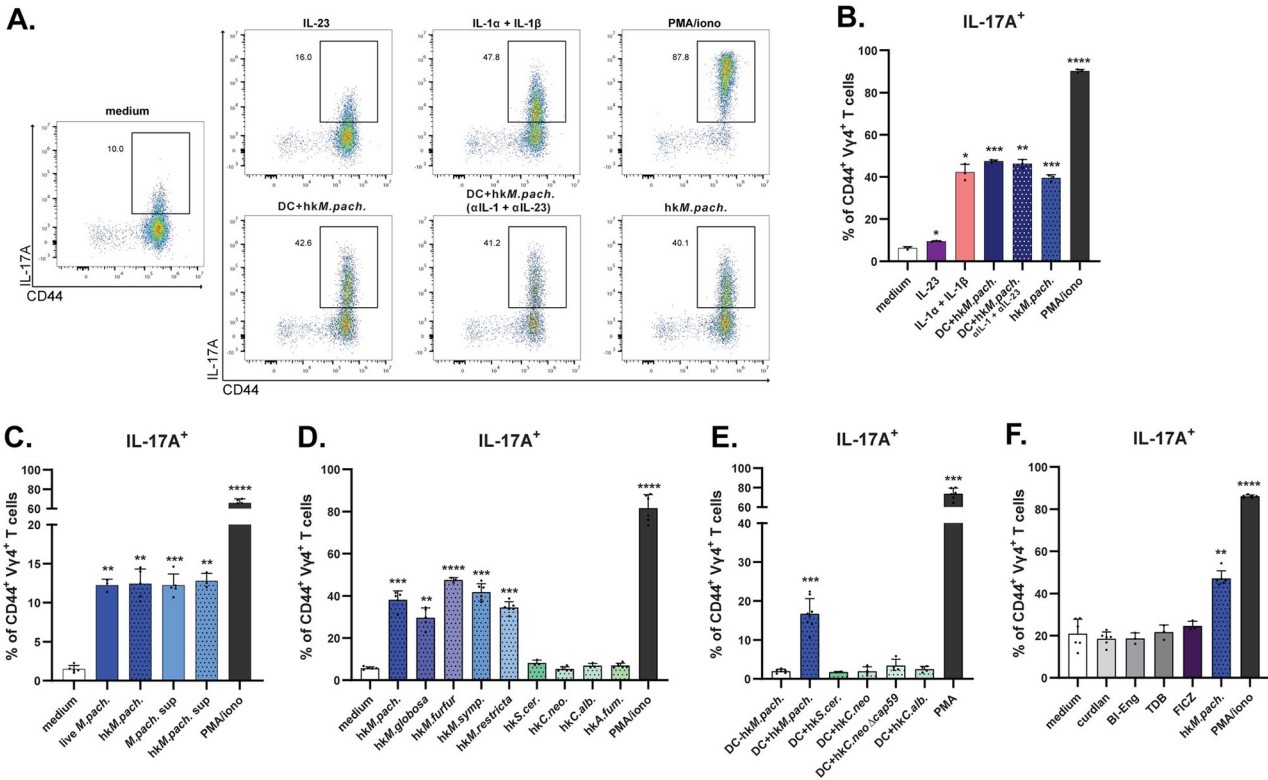

**Fig 8. γδ T cells are responding directly and specifically to *Malassezia*-derived structures. A.-F.** The ear skin of WT mice was associated with *M. pachydermatis* for 7 days and γδ T cells were FACS-purified from dLN (A-C and E-F) or used as whole LN single cell suspension (D) and re-stimulated with the indicated compounds for 5 hours before quantification of IL-17A production by CD44+ Vγ4+ γδ T cells by flow cytometry. Stimulation with cell culture medium (medium) and PMA and ionomycin (PMA/iono) was included in all experiments as a reference for the response. Representative flow cytometry plots (A) and summary graphs (B-F) showing the proportion of IL-17A+ cells among CD44+ Vγ4+ γδ T cells after the indicated stimulation conditions. hk, heat killed fungal cells; sup, supernatant from fungal cultures; *S. cer.*, *S. cerevisiae*; *C. neo.*, *C. neoformans*; *C. alb.*, *C. albicans*; *A. fum.*, *A. fumigatus*; Bl-Eng, Blastomyces Endoglucanase-2 (a prototypical Dectin-2 agonist); TDB, Trehalose-6,6-dibehenate (a prototypical Mincle agonist); FICZ, 6-Formylindolo[3,2-b]carbazole (a prototypical AhR agonist). Data are from one representative of two independent experiments (A-B) or are pooled from two (C, D, F) or three (E) independent experiments with 2–3 mice per group. Each symbol represents one mouse, the mean+SD is indicated. Statistical significance between each stimulation condition and the respective medium control was determined using paired one-way ANOVA. *p<0.05, **p<0.01, ***p<0.001, ****p<0.0001. **See also** S6 Fig.

stimulation, as previously excluded (Fig 5E–5H). However, blocking IL-1 and IL-23 during γδ T cell re-stimulation with fungus-pulsed DCs did not diminish the IL-17 response (Fig 8A and 8B). While not excluding the possible involvement of other cytokines, we next questioned the requirement for DCs during γδ T cell re-activation. For this, we assessed how IL-17 expression was impacted by the presence or absence of DCs during fungal restimulation of FACS-purified dLN γδ T cells from *M. pachydermatis*-associated mice. The proportion of Vγ4+ γδ T cells producing IL-17 was comparable between the conditions with and without DCs (Fig 8A and 8B), suggesting that the fungus can act directly and in a dose-dependent manner (S6C Fig) on Vγ4+ γδ T cells from pre-exposed mice and did not require DC-derived signals. This finding was consistent with our results from CD11c-depleted mice (Fig 5A–5D) and earlier data indicating a negligible role of TCR signalling (Fig 5E–5G). Together, these results suggest that IL-1 can elicit IL-17 production in Vγ4+ γδ T cells and contributes to primary activation of γδT17 cells upon *Malassezia* skin association, but is dispensable for re-activation of these cells by *Malassezia*-derived components.

We then further explored the ability of *M. pachydermatis* to directly induce IL-17 expression in Vγ4$^+$ γδ T cells from fungus-associated mice. Both heat-killed and live fungal cells activated Vγ4$^+$ γδ T cells with comparable potency (Fig 8C). Moreover, yeast cell-free supernatant (*M.pach*. sup) was sufficient to stimulate a robust IL-17 response (Fig 8C), indicating that the IL-17-stimulating fungal ligand is soluble and released from the fungus without need for active *de novo* synthesis. In contrast, Vγ4$^+$ γδ T cells from vehicle-treated control animals did not respond to the *Malassezia*-derived stimulus (S6D Fig) indicating that additional signals are required for initial activation *in vivo* via MyD88/IL-1- and IL-23-dependent but CLR- and TLR-independent pathways (Fig 6). The γδ T cell stimulating activity was conserved across several species of *Malassezia* (Fig 8D). However, other Basidiomycota (*C. neoformans* WT and the acapsular mutant Δcap59 [49]) or Ascomycota (*C. albicans*, *S. cerevisiae*, *A. fumigatus*) failed to elicit IL-17 production from Vγ4$^+$ γδ T cells from *Malassezia*-exposed mice, irrespective of the presence or absence of DCs (Fig 8D and 8E). To exclude that these effects were due to the different media used to culture fungi, we tested whether *C. albicans* grown in lipid-rich mDixon medium (used for culturing *Malassezia*) rather than standard YPD was able to activate γδT17 cells, but were still unable to detect IL-17 expression (S5D Fig). This result indicated that the *Malassezia*-derived Vγ4$^+$ γδ T cell ligand was not a widely conserved fungal PAMP. Accordingly, prototypical CLR agonists β-glucan (as in curdlan), Blastomyces Eng2 glycoprotein (Bl-Eng2), or mincle agonist trehalose-6,6-dibehenate (TDB) were unable to trigger IL-17 production in Vγ4$^+$ γδ T cells (Fig 8F), thus confirming the CLR-independent nature of this response (Fig 7A). In addition to Dectin-1 and TLR2, Vγ4$^+$ γδ T cells have been shown to respond to aryl hydrocarbon receptor (AhR) stimulation [5], and *Malassezia* can secrete bioactive indoles that serve as AhR agonists [17]. However, Vγ4$^+$ γδ T cells from *Malassezia*-exposed mice did not respond to AhR stimulation with the potent agonist 6-Formylindolo [3,2-b]carbazole (FICZ), precluding that the IL-17 stimulating fungal ligand was acting via AhR (Fig 8F). Together, these results identify that a *Malassezia*-derived factor unrelated to known fungal ligands can act directly on γδ T cells to elicit IL-17 release and appears remarkably restricted to the *Malassezia* genus.

## Discussion

Dermal γδ T cells are an integral part of the cutaneous immunosurveillance program. By providing IL-17 these cells mediate skin immunity and wound repair processes that are vital to maintain barrier function against constant exposure to pathogens and toxins. Here, we identify Vγ4$^+$ γδ T cells as key players governing type 17 immunity against *Malassezia*, the prevailing fungal constituent of the skin microbiome in both humans and animals. We demonstrate a dynamic and long-lasting protective ability of γδT17 cells to mediate fungal control, shed light on their mode of activation by IL-23 and IL-1 in the absence of antigen presenting cells, and highlight their direct response to conserved *Malassezia*-released factors.

Type 17 responses are a common feature of antifungal immunity in barrier tissues, including the oral and gastrointestinal mucosa which are colonized by *Candida* among other fungal genera [50]. In the oral mucosa, the IL-17 response is mediated by co-operation of diverse T cells and innate lymphoid cell subsets, especially during early time points of infection, which has been extensively probed by our group and others using experimental models of oropharyngeal candidiasis [51,52]. In a cutaneous model of *C. albicans* infection however, the antifungal IL-17 response is dominated by dermal γδ T cells [11], as we show here for *M. pachydermatis* as well as *M. furfur* and *M. sympodialis*, consistent with a specialised role for γδ T cells in this tissue environment. Remarkably, γδ T cells remain critical for *Malassezia* control over several weeks, long after αβ Th17 cells have expanded and acquired effector functions.

Therefore, other T cells can only partially compensate in this setting, contrasting with many examples in which *Tcrd*$^{-/-}$ mice display a weak phenotype because plastic αβ T cells can occupy the vacant niches [53]. Nonetheless, differential fungal clearance in *Tcrd*$^{-/-}$ vs. *Il17af*$^{-/-}$ mice indicates that at later time points cells other than γδ T cells also contribute to overall IL-17-mediated immunity to *Malassezia*.

In line with their long-lasting response against *Malassezia*, γδT17 cells adopt memory-like features as highlighted by enhanced responsiveness to a secondary infection with this fungus. γδ T cells are increasingly recognized as versatile cells and assigned features of adaptive immunity [6]. Memory γδ T cells were first reported in the context of intestinal infection with *Listeria* [54]. Shortly after, the emergence of memory-like γδ T cells was documented in the imiquimod-induced model of psoriasis in which γδ T cells were shown to exacerbate psoriaform inflammation [55,56]. In response to *Malassezia*, we observed that memory-like Vγ4$^+$ γδ T cells accumulate in the dLN and persist for several weeks after the fungus was cleared. Strikingly, the persisting γδ T cell population was associated with increased cytokine responses and enhanced fungal control upon re-exposure to the organism.

Dermal γδT17 cells comprise both resident and self-renewing cells that seed the tissues at early pre- and post-embrionic stages [8,57], as well as γδT17 cells derived from adult bone marrow that expand in the draining lymph nodes, especially under inflammatory conditions [45]. Barrier tissue-draining lymph nodes also contain CCR6$^+$ γδ T cells that are pre-committed to produce IL-17 in response to microbial encounter [58]. We did not observe any co-production of IFN-γ by those cells. While we did not dissect the origin of the γδ T cells involved in the antifungal response, we observed that a significant proportion of the γδT17 population in *Malassezia*-exposed skin depends on activation in the dLN. This is supported by pronounced γδT17 cell expansion and IL-17 production in the lymph nodes, as well as the reduced number of proliferating dermal γδ T cells after pharmacological blockade of lymphocyte egress from the dLN. These results are consistent with the observed pattern of tissue-homing receptor expression by the IL-17$^+$ γδ T cell subset in the dLN, while all dermal γδ T cells were uniformly CCR6 and CXCR6 positive. Together, these data support an important contribution of γδ T cell trafficking from the dLNs to antifungal responses in the colonized skin.

Despite their emerging adaptive features, γδT17 cells are often referred to as 'innate-like', since their effector functions appear pre-determined in the thymus. However, these cells must still undergo activation for their full effector functions to be induced [4]. The signals involved in γδT17 activation are versatile [29,59], and mostly independent of cognate antigen [44,45]. Accordingly, *Malassezia* stimulation of γδ T cells cells does not substantially engage the TCR, and activation of γδT17 cells in the *Malassezia*-exposed skin depended on IL-1 and IL-23 rather than CD11c$^+$ antigen-presenting cells. *Malassezia* was previously shown to stimulate production of IL-1β in a Syk- and NLRP3-dependent manner [32,60–62]. Whether this applies to IL-1 family cytokine induction in the *Malassezia*-exposed skin *in vivo* remains unclear. We identified and confirmed Ly6G$^+$/Ly6C$^+$ myeloid cells as a major early source of this cytokine during *Malassezia* colonization, while excluding TLRs, CLRs, and Card9-dependent pathways which were redundant for γδT17 activation. Other fungi such as *C. albicans* and *A. fumigatus* elicit IL-1 family cytokines in a CLR-independent manner by triggering host cell damage [63,64], although *Malassezia* is not a significant inducer of cellular damage in our hands. *Malassezia* triggering of IL-23 expression may involve skin sensory neuron detection of the fungus, as was shown for *C. albicans* in a model of cutaneous infection [11]. Whether and how neuro-immunological crosstalk is implicated in the type 17 response to *Malassezia* will be an interesting avenue to explore in future.

In addition to cytokine activation, γδ T cells may also detect fungal PAMPs directly via PRRs including TLR2 and dectin-1, since these cells have been shown to release IL-17 in

response to Pam$_3$CSK$_4$ and curdlan stimulation [5]. However, in the *Malassezia*-colonized skin, deficiency in TLRs, CLRs, or Card9 did not impact γδT17 activation, and CLR ligands did not elicit IL-17 expression by Vγ4$^+$ γδ T cells isolated from colonized animals. This is consistent with our finding that none of the alternative fungal genera tested were able to induce a comparable response. At the same time, the γδT17-stimulating activity of *Malassezia* appeared to be conserved across numerous *Malassezia* species, including *M. restricta* and *M. globosa*, the most common colonizers of human skin [14]. Although the molecular identity of the γδ T cell-stimulating activity in *Malassezia* remains to be determined, its restriction to a single fungal genus is remarkable. Given that multiple *Malassezia* species can colonize human skin, future studies will need to assess whether reactivity to this fungal agonist is conserved in human γδ T cell subsets, including possible effects on cell trafficking and localisation, as well as effector functions within cutaneous tissues. Indeed, *Malassezia* has recently been identified in tumors and been associated with significantly lower overall survival rates, especially in breast cancer. [65,66] Since γδ T cells are key players in tumor immunity and emerging tools in immunotherapy, [67, 68] investigating the role of *Malassezia*-reactive γδ T cells in tumorigenesis may reveal how the presence of this fungus is linked with cancer progression and patient survival.

In addition to direct fungal responsiveness, we observed a striking longevity of murine γδT17-mediated cutaneous protection against *Malassezia*. This finding raises the question of whether naturally colonized skin is subject to tonic γδ T cell stimulation by the constant presence of *Malassezia*, akin to the 'normality sensing' recognition of butyrophilin-like molecules [69, 70]. We thus propose that initial activation of murine γδT17 by IL-23 and IL-1 family cytokines is followed by direct fungal sensing to facilitate sustained type 17 immunity, thereby mediating surveillance of potentially tissue-disruptive challenges and maintaining commensalism.

Our work demonstrates that γδT17 cells contribute to host protection against *Malassezia* in normal skin. While *Malassezia* is a skin commensal organism, this fungus is also associated with allergic and other chronic and recurrent inflammatory skin disorders. In future it will be interesting to assess the role of γδT17 cells in inflamed skin to understand whether they exert disease-promoting or host-beneficial effects in diverse skin conditions. Of note, exacerbated γδ T cell expansion and activation is a common feature of skin inflammation [2]. Moreover, *Malassezia* colonization exacerbates IL-17-dependent inflammation in barrier-disrupted murine skin, although it remains unclear whether this process is mediated by γδ T cells [24]. Whether or not the fungal ligand/s that drive IL-17 secretion by γδ T cells are differentially expressed by *Malassezia* in healthy versus diseased skin also remains an open question.

## Methods

### Ethics statement

All mouse experiments in this study were conducted in strict accordance with the guidelines of the Swiss Animals Protection Law and were performed under the protocols approved by the Veterinary office of the Canton Zurich, Switzerland (license number 168/2018 and 142/2021). All efforts were made to minimize suffering and ensure the highest ethical and humane standards according to the 3R principles [71].

### Animals

WT C57BL/6j mice were purchased by Janvier Elevage. *Il17af-/-* [57], *Tcrd-/-* [26], *Card9-/-* [24], *MyD88-/-* [48], *Clec4n-/-* [36] and *Clec7a-/-* [35] (kindly provided by Gordon Brown, University of Exeter, UK), *Il23r-/-* [72] (kindly provided by Burkhard Becher, University of Zürich), *Il23a-/-* [73], *Nur77*$^{GFP}$ [31] (kindly provided by Onur Boyman, University Hospital

Zurich, Switzerland), *Tlr23479-/-* [39] (kindly provided by Thorsten Buch, University of Zürich, Switzerland), and *Il17a^{Cre} R26R^{eYFP}* mice [25] were bred at the Institute of Laboratory Animals Science (LASC, University of Zurich, Zurich, Switzerland). All mice were on the C57BL/6 background. The animals were kept in specific pathogen-free conditions and used at 8–14 weeks of age in sex- and age-matched groups. Female as well as male mice were used for experiments.

## Generation of chimeric mice

C57BL/6 recipient mice of 6–8 weeks were irradiated twice with a dose of 5.5 Gy at an interval of 12 h. CD11c-DTR+ and CD11c-DTR- littermate control mice [30], or *Clec4e-/-* [37] and WT C57BL/6 controls mice (kindly obtained from David Sancho, CNIC, Spain), respectively, served as bone marrow donors. The donor bone marrow was injected in the tail vain of five recipient mice per donor mouse, 6 h after the second irradiation. Mice were treated with Borgal (MSD Animal Health GmbH) p.o. for the first 2 weeks of an 8-week reconstitution period. For depletion in CD11c-DTR+/- chimeras, both groups were treated with 1 µg diphtheria toxin i.p. one day prior to *Malassezia* colonization and at the day of colonization. Depletion of CD11c$^+$ cells was confirmed by flow cytometry analysis.

## Fungal strains

*M. pachydermatis* strain ATCC 14522 (CBS 1879, [74]) and *M. globosa* strain MYA-4612 (CBS7966) were purchased from ATCC. *M. sympodialis* strain ATCC 42132 [75] and *M. furfur* strain JPLK23 (CBS 14141, [74]) were obtained from Joseph Heitman (Duke University Medical Center, Durham, NC). *M. restricta* from skin of a healthy subject was obtained from Philipp Bosshard (University Hospital Zurich, Switzerland). All *Malassezia* strains were grown in mDixon medium at 30°C and 180 rpm for 2–3 days. *C. albicans* strain SC5314 [76] and *S. cerevisiae* strain BY4741 (obtained from Vikram Panse, University of Zürich, Switzerland) were grown in YPD at 30°C and 180 rpm overnight. As a control, *C. albicans* was also grown in mDixon medium at 30°C and 180 rpm for 2 days. An azole-sensitive isolate of *Aspergillus fumigatus* (kindly provided by Dominique Sanglard, Centre Hospitalier Universitaire Vaudois, Switzerland) was grown on malt yeast extract (MYA) agar at 42°C for 2–3 days. Heat-killed preparations of *C. neoformans* strain H99 [77] and Δcap59 (from the Jenny Lodge laboratory) were kindly provided by Andrew Alspaugh (Duke University Medical Center, Durham, NC). Heat-killing was achieved by incubating fungal suspensions at $OD_{600} = 1$/ml in PBS for 45 min at 85°C. *M. pachydermatis* supernatant was obtained by culturing in mDixon for 2 days. Fungal cells were washed twice in PBS and adjused to 1 $OD_{A600}$ in 3ml ($\approx$ 5 x 10$^6$ yeast cells in 3 ml) in cRPMI and incubated overnight at 30°C and 180 rpm. Supernatants were obtained after centrifugation (5 min, 880xg) and sterile filtration with a 0.22 µm filter.

## Epicutaneous association of mice with *Malassezia*

Epicutaneous colonistaion of the mouse ear skin was performed as described previously [24,78]. In short, *Malassezia* cells were washed with PBS and suspended in commercially available native olive oil at a density of 10 $OD_{A600}$/ml. 100 µL suspension (corresponding to 1 $OD_{A600}$ $\approx$ 5 x 10$^6$ yeast cells) was topically applied onto the dorsal ear skin while mice were anaesthetized. Ear thickness was measured prior and during the course of infection using the Oditest S0247 0–5 mm measurement device (Kroeplin). Animals treated with olive oil (vehicle) and infected animals were kept separately to avoid transmission. For determining the fungal loads in the skin, tissue was homogenzied in water supplemented with 0.05% Nonidet P40

(AxonLab), using a Tissue Lyser (Qiagen), and plated on mDixon agar for incubation at 30˚C for 3 to 4 days.

## FTY720 treatment

Mice were treated with 5 μg/ml FTY720 (Selleckchem) or 0.3 μl ethanol control per ml in the drinking water starting 1 day prior to colonization and during the entire duration of the experiment [79]. The supplemented water was exchanged on day 4.

## Neutrophil depletion

Mice were treated intraperitoneally with 1 μg/ml (125 μl/injection) anti-Gr1 (NIMP-R14) starting 1 day prior to colonization and treatment continued daily until one day prior to at the experimental endpoint. Control mice were injected with equal amounts of PBS.

## Histology

Mouse tissue was fixed in 4% PBS-buffered paraformaldehyde overnight and embedded in paraffin. Sagittal sections (9μm) were stained with hematoxylin and eosin and mounted with Pertex (Biosystem, Switzerland) according to standard protocols. All images were acquired with a digital slide scanner (NanoZoomer 2.0-HT, Hamamatsu) and analyzed with NDP view2 (Hamamatsu).

## RNA extraction and RT-qPCR

Isolation of total RNA from murine ear skin was performed using TRI reagent (Sigma-Aldrich) according to the manufacturer's protocol. cDNA was generated by RevertAid reverse transcriptase (ThermoFisher). Quantitative PCR was performed using SYBR green (Roche) and a QuantStudio 7 Flex instrument (Life Technologies). The primers used for qPCR were *Actb* forward 5'-CCCTGAAGTACCCCATTGAAC-3', *Actb* reverse 5'-CTTTTCACGGTTGGCCTTAG-3'; *Il17a* forward 5'-GCTCCAGAAGGCCCTCAGA-3', *Il17a* reverse 5'-AGCTTTCCCTCCGCATTGA-3'; *Il17f* forward 5'-GAGGATAACACTGT-GAGAGTTGAC-3', *Il17f* reverse 5'-GAGTTCATGGTGCTGTCTTCC-3'; *Il23a* forward 5'-CCAGCAGCTCTCTCGGAATC-3', *Il23a* reverse 5'- TCATATGTCCCGCTGGTGC-3'; *Il1b* forward 5'- GAGCTGAAAGCTCTCCACCTC-3', *Il1b* reverse 5'- CTTTCCTTTGAGGCC-CAAGGC-3'. All qPCR reactions were performed in duplicates, and the relative expression (rel. expr.) of each gene was determined after normalization to *Actb* transcript levels.

## Isolation of skin and lymph node cells

For digestion of total ear skin, mouse ears were cut into small pieces and transferred into $Ca^{2+}$- and $Mg^{2+}$-free Hank's medium (Life Technology) supplemented with Liberase TM (0.15 mg/mL, Roche) and DNase I (0.12 mg/mL, Sigma-Aldrich) and incubated for 45 min at 37˚C. Ear-draining lymph nodes (dLN) were digested with DNAse I (2.4mg/ml Sigma-Aldrich) and Collagenase I (2.4mg/ml, Roche) in PBS for 15 min at 37˚C. Both cell suspensions were filtered through a 70 μm cell strainer (Falcon) and rinsed with PBS supplemented with 5 mM EDTA (Life Technologies), 1% fetal calf serum and 0.02% $NaN_3$.

## *Ex vivo* T cell re-stimulation

For *in vitro* re-stimulation of γδ T cells, skin cell suspensions were incubated in a U-bottom 96-well plate (1/4 ear per well) with cRPMI medium (RPMI supplemented with 10% fetal calf serum, HEPES, sodium pyruvate, non-essential amino acids, β-mercaptoethanol, Penicillin

and Streptomycin) with phorbol 12-myristate 13-acetate (PMA, 50 ng/ml, Sigma-Aldrich) and ionomycin (500 ng/ml, Sigma-Aldrich) for 5 h at 37 ˚C in the presence of Brefeldin A (10 μg/ml). $10^6$ lymph node cells or $2 \times 10^4$ FACS-purified γδ T cells per well of a U-bottom 96-well plate were co-cultured with $1 \times 10^5$ DC1940 cells [80] that were pulsed with $2.5 \times 10^5$ heat-killed fungal cells. Brefeldin A (10 mg/ml, Sigma-Aldrich) was added for the last 5 hours. After stimulation, cells were surface stained, fixed and permeabilized with Cytofix/Cytoperm (BD Biosciences) and then stained for cytokine expression. In some experiments, lymph node cells or FACS sorted γδ T cells were stimulated in absence of DC1940 cells with 20 μl of $OD_{A600}$1 adjusted heat-killed fungal extracts, 100 μl fungal supernatants, 100 ng/ml recombinant mouse IL-23 (R&D Systems), 50 ng/ml recombinant IL-1α (Peprotech) and 50 ng/ml recombinant IL-1β (Peprotech), 100 μg/ml curdlan [81], 1.25 ng/ml Blastomyces endoglucanase-2 (Bl-Eng2, kindly provided by Marcel Wüthrich, University of Wisconsin, Madison, WI [82]), 20 μg/ml trehalose-6,6-dibehenate (TDB [83], Invivogen), or 1 μM 6-formylindolo[3,2-*b*]carbazole (FICZ [84], Invivogen). Cytokine blockade was achieved using IL-1 receptor A antagonist anakinra (10 μg/ml, Kineret, Sobi, Germany) and anti-mouse IL-12p40 (2 μg/ml, clone C17.8, BioXcell).

### Flow cytometry

Single cell suspensions were stained with antibodies as listed in S1 Table. LIVE/DEAD Fixable Near IR stain (Life Technologies) was used for exclusion of dead cells. After surface staining, cells were fixed and for subsequent intracellular staining permeabilized with Cytofix/Cytoperm (BD Biosciences) and stained in Perm/Wash buffer (BD Bioscience) for cytokines, as appropriate. All staining steps were carried out on ice. Cells were acquired on a Spectral Analyzer SP6800 (Sony) or on a CytoFLEX S (Beckman Coulter), and the data were analyzed with FlowJo software (FlowJo LLC). The gating of the flow cytometric data was performed according to the guidelines for the use of flow cytometry and cell sorting in immunological studies [85], including pre-gating on viable and single cells for analysis. Absolute cell numbers were calculated based on a defined number of counting beads (BD Bioscience, Calibrite Beads) that were added to the samples before flow cytometric acquisition. For assessing TCR activation in γδ T cells from Nur77$^{GFP}$ reporter mice, the MFI of GFP expression was determined in lymph node cells that were stimulated and stained as described above.

### Sorting of γδ T cells

Single cell suspensions from dLN were stained for TCRβ, CD4, CD8, CD19, CD11b, I-A/I-E (all FITC-conjugated) as described above. Cells were washed and resuspended in cRPMI and kept at 4˚C for sorting on a FACSAria III (BD Bioscience). γδ T cells were sorted based on scatter, singlets, and negativity for FITC according to an unstained control. Sorting resulted in a purity of 80% TCRγδ$^+$ cells on average while FITC$^+$ negatively-selected cells remained below 1%. Cells were collected in cRPMI at 4˚C for T cell re-stimulation assays.

### Supporting information

**S1 Fig. (related to Fig 1) Cutaneous immunity against *Malassezia* depends on IL-17A and IL-17F cytokines.** The ear skin of wild type (WT) and *Il17af*$^{-/-}$ mice was associated with *M. pachydermatis* or treated with olive oil (vehicle control). **A**. Hematoxylin and eosin-stained WT ear tissue section on day 2 after *M. pachydermatis* association shown in Fig 1C and magnification of the indicated area. **B**. Gating strategy for neutrophils (Ly6G$^{hi}$ CD11b$^+$) among viable CD45$^+$ cells in the ear skin of WT and *Il17af*$^{-/-}$ mice 4 days after association with *M.*

*pachydermatis*.
(TIF)

**S2 Fig. (related to Fig 2) Vγ4⁺ γδ T cells are the main IL-17A producers in *Malassezia*-associated skin. A-D.** The ear skin of *Il17a*$^{Cre}$ *R26R*$^{eYFP}$ fate reporter mice was associated with *M. pachydermatis* and the IL-17-expressing cellular subsets were assessed by flow cytometry analysis of eYFP reporter expression. **A.** Gating strategy for identifying the eYFP⁺ cellular subsets among viable CD45⁺ CD11b⁻ single cells. eYFP expression levels are indicated by a colour scale. **B.-C.** Quantification of eYFP⁺ cells among overall skin CD90⁺ cells (B) or Vγ4⁺ TCRγδ⁺ CD90⁺ T cells (C) at the indicated time points (dpi) and in uninfected control animals (vehicle). **D.** Proportion of eYFP⁺ cells among the indicated cell populations in the ear skin at the indicated time points (dpi) and in uninfected control animals (vehicle). Data in A—D are from two independent experiments per time point with 2–4 mice per group; each symbol represents one mouse; the median of each group is indicated. **E.** IL-17A⁺ cells after the indicated re-stimulation in *M. pachydermatis*-associated or vehicle control mice on day 7. Data are pooled from two independent experiments with 3–4 mice per group. Each data point represents one mouse, the median is indicated. **F.-I.** WT mice were associated with *M. furfur* (F, G) or *M. sympodialis* (H, I) for 7 days and γδ T cells were quantified in ear skin and dLN. Quantification of IL-17A⁺ cells among Vγ4⁺ TCRγδ⁺ CD90⁺ T cells (F) or among TCRγδ⁺ CD90⁺ T cells (H) after ex vivo re-stimulation with PMA and ionomycin (skin) or with heat-killed *Malassezia* spp.-pulsed DCs (dLN). Proportion of IL-17A⁺ cells among the indicated cell populations in the ear skin and dLN (G, I). Data in F-I are from one experiment with 3 mice per group. **K.-L.** Phenotype of IL-17A⁺ Vγ4⁺ (dark blue, solid line) and IL-17A⁻ Vγ4⁺ γδ T cells (light blue, dashed line) in dLN (K) and ear skin (L) after restimulation with hk*M.pach.*-pulsed DCs or PMA/ionomycin, respectively. Histograms are from three concatenated samples. Data are from three (K) or two (L) independent experiments with 3–5 mice per group. Statistical significance (B, C, E) was determined using one-way ANOVA. ***$p<0.001$, ****$p<0.0001$. (TIF)

**S3 Fig. (related to Fig 4) A pool of γδ T cells in lymph nodes supports long-term protection against *Malassezia*. A-E.** WT mice were associated with *M. pachydermatis* (*M.pach.*) once or twice or treated with olive oil (vehicle control) as in Fig 4A–4E. Gating strategy for quantifying IL-17 producing TCRγδ⁻ CD4⁺ T cells and Vγ4⁺ TCRγδ⁺ T cells (A). Increase in ear thickness (B). Total numbers of neutrophils (C) and TCRγδ⁻ IL-17⁺ CD4⁺ T cells in the skin (D). Total numbers of *M. pachydermatis*-reactive IL-17A⁺ TCRγδ⁻ CD4⁺ T cells in dLN (E). Data in B-D are compiled from two independent experiments, and in E are from one representative of two independent experiments with 2–4 mice per group. Each symbol represents one mouse. The mean+SEM per group is indicated in B, the median per group is indicated in C-E. **F.-I.** WT mice were associated with *M. pachydermatis* and treated with FTY720 or EtOH solvent control as in Fig 4F–4I, but analyzed at 4 dpi instead of 7 dpi. Total numbers of Ki-67⁺ CD44⁺ γδ T cells in dLN (F) and ear skin (G), increase in ear thickness (H), and skin fungal load (I). Data are from one representative of two (F-G) or pooled from two (H-I) independent experiments. Each symbol represents one mouse; the median per group is indicated. Statistical significance was determined using one-way ANOVA (B-G), two-way ANOVA (H), or Student's *t* test (I). *$p<0.05$, **$p<0.01$, ***$p<0.001$, ****$p<0.0001$. (TIF)

**S4 Fig. (related to Fig 6). γδ T cells are activated independently of C-type lectin and Toll-like receptor signalling in response to *Malassezia*.** The ear skin of *Clec7a*$^{-/-}$ and *Clec7a*$^{+/-}$ littermate control mice (A-C, K), *Clec4n*$^{-/-}$ and *Clec4n*$^{+/-}$ littermate control mice (D-F, L),

irradiation chimeras reconstituted with *Clec4e*[-/-] or WT control bone marrow (G-I, M), *Card9*[-/-] and *Card9*[+/-] littermate control mice (N) or *Tlr23479*[-/-] and control mice (O) was associated with *M. pachydermatis*. After 7 days, γδ T cells in skin and dLN were quantified (A, D, G) and analyzed for IL-17 production (B, E, H). Skin fungal load (CFU) (C, F, I). Increase in skin ear thickness (K-O). Data are pooled from two (A-F, K, L, O) or three (G-I, M, N) independent experiments with 3–5 mice per group. Each symbol represents one mouse. The median of each group is indicated. DL, detection limit. Statistical significance was determined using unpaired Student's t-test. **$p < 0.01$.
(TIF)

**S5 Fig. (related to Fig 7). The antifungal γδ T cell response depends on IL-23 and IL-1 family cytokine signalling. A.-B.** The ear skin of *Il23a*[-/-] and C57BL/6 control mice was associated with *M. pachydermatis* for 7 days, when skin γδ T cells were quantified (A) and the fungal load (CFU) was assessed in the skin (B). **C.-E.** The ear skin of *Il23r*[-/-] and *Il23r*[+/-] littermate control (C, E) and of *MyD88*[-/-] and *MyD88*[+/-] littermate control mice (D) was associated with *M. pachydermatis* or vehicle treated for 7 days. Ear thickness kinetics (C-D) and skin neutrophils were quantified. Data are pooled from three (D) independent experiments with 2–5 mice per group each. The mean +/- SEM of each group is shown in C-D. In A-B and E, each symbol represents one mouse and the median of each group is indicated. Statistical significans was determined using unpaired Student's *t* test (A-B) or two-way ANOVA (C-E). *$p < 0.05$, ***$p < 0.001$, ****$p < 0.0001$.
(TIF)

**S6 Fig. (related to Fig 8). γδ T cells respond directly and specifically to *Malassezia*-derived structures. A.-D.** γδ T cells were purified by FACS from dLN (A-C) or used as whole dLN suspension (D) of WT mice that were associated with *M. pachydermatis* (A-D) or treated with olive oil (vehicle, D) for 7 days and re-stimulated with the indicated compounds for 5 hours before quantification of IL-17A production by CD44[+] Vγ4[+] γδ T cells by flow cytometry as in Fig 7. Purity of FACS-sorted TCRγδ[+] cells. shown by representative plots (A) and in a summary graph of data pooled from all experiments in Figs 7 and S5D (B). Re-stimulation with heat-killed *M. pachydermatis* (hk*M.pach.*) at different concentrations or with hk *C. albicans* grown in mDixon instead of YPD medium (C), or with hk*M.pach.* or *M. pachydermatis* supernatant (*M.pach.* sup; D). Stimulation with cell culture medium (medium) and PMA and ionomycin (PMA/iono) were included as reference for the response. In B, each symbol represents one mouse and mean+/-SEM is indicated. In C and D, each symbol represents one mouse, the mean+SD is indicated. Statistical significance between each stimulation condition and the respective medium control was determined using paired one-way ANOVA. *$p < 0.05$, **$p < 0.01$, ***$p < 0.001$, ****$p < 0.0001$.
(TIF)

**S1 Table. List of all antibodies used in this study.**
(DOCX)

# Acknowledgments

The authors would like to thank Gordon Brown, Thorsten Buch and Onur Boyman for mice; David Sancho for bone marrow; Joseph Heitman, Andrew Alspaugh, Philipp Bosshard, Dominique Sanglard and Vikram Panse for fungal strains; Marcel Wüthrich for Dectin-2 agonist Bl-Eng2; the staff of the Laboratory Animal Service Center of University of Zürich for animal

husbandry; staff of the Laboratory for Animal Model Pathology of University of Zürich for histology; members of the LeibundGut-lab for helpful advice and discussions.

## Author Contributions

**Conceptualization:** Fiorella Ruchti, Salomé LeibundGut-Landmann.

**Formal analysis:** Fiorella Ruchti, Meret Tuor, Liya Mathew.

**Funding acquisition:** Neil E McCarthy, Salomé LeibundGut-Landmann.

**Investigation:** Fiorella Ruchti, Meret Tuor, Liya Mathew.

**Methodology:** Fiorella Ruchti.

**Project administration:** Salomé LeibundGut-Landmann.

**Supervision:** Neil E McCarthy, Salomé LeibundGut-Landmann.

**Validation:** Fiorella Ruchti, Neil E McCarthy, Salomé LeibundGut-Landmann.

**Visualization:** Fiorella Ruchti.

**Writing – original draft:** Fiorella Ruchti, Salomé LeibundGut-Landmann.

**Writing – review & editing:** Fiorella Ruchti, Meret Tuor, Liya Mathew, Neil E McCarthy, Salomé LeibundGut-Landmann.

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
