## [Decision Letter · Decision Letter 0]

6 Oct 2023

Dear Dr. LeibundGut-Landmann,

Thank you very much for submitting your manuscript "γδ T cells respond directly and selectively to the skin commensal yeast Malassezia for IL-17-dependent fungal control." for consideration at PLOS Pathogens. As with all papers reviewed by the journal, your manuscript was reviewed by members of the editorial board and by several independent reviewers. The reviewers appreciated the attention to an important topic. Based on the reviews, we are likely to accept this manuscript for publication, providing that you modify the manuscript according to the review recommendations.

Please carefully review concerns from reviewers 1 and 3. Although multiple additional experimental questions are posed, the reviewers and editors also appreciate the significant strengths of the work presented. We understand that adding all the suggested experiments will be difficult, please provide a detailed response to all points, add any data you might have and clarify the number of mice used in each experiment. In addition to providing a detail rebuttal letter, please also edit results and discussion to include mention of limitations of the work and consider incorporating answers to the reviewer's questions especially as it relates to differences between mouse and human gamma delta T cells and how much of the work can be confidently extrapolated to responses in humans.

Sincerely,

Amariliz Rivera

Academic Editor

PLOS Pathogens

Alex Andrianopoulos

Section Editor

PLOS Pathogens

Kasturi Haldar

Editor-in-Chief

PLOS Pathogens

orcid.org/0000-0001-5065-158X

Michael Malim

Editor-in-Chief

PLOS Pathogens

orcid.org/0000-0002-7699-2064

Please carefully review concerns from reviewers 1 and 3. Although multiple additional experimental questions are posed, the reviewers and editors also appreciate the significant strengths of the work presented. We understand that adding all the suggested experiments will be difficult, please provide a detailed response to all points, add any data you might have and clarify the number of mice used in each experiment. In addition to providing a detail rebuttal letter, please also edit results and discussion to include mention of limitations of the work and consider incorporating answers to the reviewer's questions especially as it relates to differences between mouse and human gamma delta T cells and how much of the work can be confidently extrapolated to responses in humans.

Reviewer Comments (if any, and for reference):

Reviewer's Responses to Questions

**Part I - Summary**

Reviewer #1: The manuscript by Ruchti et al provides an interesting and informative dissection of the relevance of γδ T cells to Malassezia spp. infection in mice and immunological mechanisms involved. There are several strong aspects of the study, the core one being the solid evidence it provides for an important role for γδ T cells in IL-17-mediated responses to Malassezia spp infection. Of particular interest are the findings that:

- γδ T cells suppress fungal load, and can’t be compensated (at least for weeks) by the αβ T cell compartment

- that γδ T cells are key producers of IL-17 (specifically the Vγ4 subset previously associated with IL-17 production), and that this depends on both IL-23 and IL-1 cytokines.

- that γδ functionality in Malassezia spp infection appears to be independent of TCR and DCs, and also of C-type-lectins TLRs.

- that there is expansion/accumulation of Vγ4 T cells in the draining lymph nodes, and apparent trafficking to the site of infection in the skin

- that there are memory-like aspects to the role of Vγ4 T cells in Malassezia spp infection. Specifically, there is a persistently expanded population of Vγ4 T cells in the lymph node, enhanced control of fungal load after upon secondary exposure, and a direct response of Malassezia spp-exposed Vγ4 T cells to Malassezia-derived components.

- that the aforementioned response to Malassezia-derived components is genus-specific, but c-type-lectin/TLR/AhR-independent, and also involves a soluble factor

Overall, this enhances substantially understanding of immunity to this important fungal commensal in mice, and includes some immunologically novel and intriguing elements concerning memory and direct pathogen sensing.

In general the study appears to be well executed and it is also well written, with a substantial and comprehensible introduction and Discussion.

As outlined below, the major weakness relates to the human aspect of the study, which warrants both editorial changes and some additional experiments.

Reviewer #2: Manuscript "gdT cell respond directly and selectively to the skin commensal yeast Malassezia for IL-17-dependent fungal control by Ruchti et al. provides novel mechanistic insight into the induction of Th17 immunosurveillance of skin commensal colonization that has significant implication for cutaneous health. This is well organized and designed study with various models used to test hypothesis. Ruchti et al extensively studied antifungal immunity in healthy mice and report that IL-17 -producing T cells represent homeostatic response against Malassezia. The only question that remains to be answered is weather fungal ligands that drive IL-17 secretion by gd T cells are differentially expressed by Malassezia in healthy versus diseased skin, as suggested by authors.

Reviewer #3: In the manuscript entitled “γδ T cells respond directly and selectively to the skin commensal yeast Malassezia for IL-17-dependent fungal control”, Ruchti et al. aim to determine the cellular and molecular mediators of Malassezia-induced type 17 immunity. Previous work in the field identified IL-17 and IL-23 (and specifically Th17 CD4+ cells) as important mediators of protection from Malassezia skin infection. While previous work also demonstrated involvement of gamma delta T cells in protection from Malassezia infection pathology, the role of gd T cells in Malassezia infection is not well understood. This is an important topic as skin infections with Malassezia are prominent in the population, and a better understanding of the host response is needed for this understudied fungal commensal/pathogen. In this manuscript the authors show the importance of IL-17 in the response to and control of M. pach and identify Vg4+ dermal gd T cells as a player in the IL-17 response to M.pach infection. Importantly, their results also demonstrate a role for memory-like gd Tcells in providing control of M. pach infection and that this response is conserved across Malassezia species. Overall, this manuscript characterizes both cellular and molecular mechanisms of the IL-17 gd T cell response during M. pach infection. I believe that this study would be of interest to the readers of this journal and the greater scientific community. However, there are a few major/minor items that need to be addressed in order to support the conclusions.

**Part II – Major Issues: Key Experiments Required for Acceptance**

Reviewer #1: The major weakness relates to the ‘human angle’ to Malassezia infection, which is insubstantial, from both a theoretica/editorial and experimental standpoint.

Experimental aspects

The human experiments at the end of the manuscript read as very much an after-thought, allowing the authors to claim that the results also apply to humans. However, there are only very limited data and experiments included, which is a shame and problematic given the current claims that are made. If at all possible, this human aspect should be expanded.

a) Unlike the mouse experiments, some of which involved isolation of Vγ4 T cells, the human PBMC assay is quite a blunt tool to assess mechanism of Malassezia-mediated effects, as multiple immune subsets will be present, and therefore any γδ T cell activation may well be indirect and downstream of initial effects on other immune cells. Experiments purifying γδ T cells and incubating them with Malassezia would determine if the effects are direct or indirect.

b) The current data are quite limited: co-culture does not appear to result in significant differences in the % of live cells that are γδ T cells, and also it is unclear from the current experiments which subset of human γδ T cells express activation markers. From the percentages involved, it appears most likely to be the numerically dominant Vγ9Vδ2 T cell subset, but that could have been easily confirmed and is potentially relevant to understanding expression profiles of putative receptors for Malassezia components. This would be a priority to investigate.

c) In these human assays, induction of cytokine production by human γδ T cells is not addressed but is obviously of interest. Also, IL-17 production by human γδ T cells is very uncommon indeed, and it seems more likely the mechanisms of any human γδ response to Malassezia might therefore be very different.

d) Also, it is unclear if the activation induced by Malassezia (assessed here by HLA-DR expression) is restricted to γδ T cells or if other immune cells within PBMC are also activated (eg αβ T cells, NK cells). This could be easily addressed and would add to understanding of the PBMC assay.

Reviewer #2: This is a very well-designed study and do not have any comments or questions. I recommend this paper for acceptance for publication.

Reviewer #3: 1) One concern for this manuscript is assigning a direct role to the IL-17 gd T cells in providing protection from M.pach infection. Specifically, an example of this is the FTY720 experiments. FTY720 blocks egress of all lymphocytes from the lymphoid tissue, including the CD4 TH17 cells. How can the role of the CD4 TH17 cells be differentiated from the IL-17 gd Tcells in protection from M.pach infection in the FTY720 experiments if both are blocked from leaving the dLN? The authors show that there is a higher number of gd T cells in the dLN at 44 days, but this does not rule out the TH17 cells from being able to expand that are in the dLN and aid in the protective responses.

An adoptive transfer of the primary infected gd Tcells into a naïve mouse could allow the authors to determine if the gd Tcells are sufficient to provide protection from M.pach infection and could help in differentiating their role compared to TH17 cells.

Also, line 195 states that IL-17 production was increased in gd Tcells in response to fungal re-exposure (4D), but does not mention that a very similar increase in IL-17 was also found in the CD4+ cells (FigS3E). Please add a description of this result into the main text.

2) Why are ear thickness/inflammation only shown for some knockout mice and not for others? For example, the FTY720-blockade experiment (Fig4), did this blockade reduce the inflammation as found in the Tcdr-/- and IL-17af-/-? What happened to the ear thickness of the CD11c DTR mice (Fig 5)? Same for figures 6 and S4. Was the inflammation of the ear the same between the control and knockout mice? If the inflammation is tied to the actions of the IL-17 gd Tcells (which are unaltered in most of these mice), then the thickness should not be affected. If it is, that should be discussed. These data are important for interpreting how inflammation is related to the level of IL-17 gd Tcells.

3) Some of the experiments, specifically ones involving IL23R+/- mice (Fig 8D,E), only have 2 animals in some of the groups. These should be repeated to reach a sufficient power for interpretation. Additionally, the figure legend says that data are pooled from 2 experiments with n=2-5 mice per group. Were the vehicle controls only done once with 2 mice?

**Part III – Minor Issues: Editorial and Data Presentation Modifications**

Reviewer #1: Theoretical/Editorial aspects

(i) The study purports to be relevant to human infection, but the Introduction is exclusively focussed on mice, with essentially no references to human γδ T cells. Some comments are incorrect when applied to the human γδ T cell compartment, such as in relation to innate pre-programming, which is the case for Vγ9Vδ2 T cells, but not for Vδ2neg T cells. If aiming to be relevant to the human compartment, then references to cover both innate-like Vγ9Vδ2 T cells, and adaptive-like Vd2neg and Vγ9negVd2 T cells, should be included. Some relevant references would be those by the Willcox group, chiefly Davey et al, 2017 (PMID 28248310), Davey et al, 2018a and 2018b (PMID 29720665 and 29680462); and Willcox CR et al, 2020 (PMID 33084045). If not aiming to apply to human γδ T cells, then the introduction text should make it clear the authors are talking exclusively about the mouse γδ T cell compartment. Given the inclusion of human data later in the manuscript it would seem logical to broaden the scope to cover human γδ T cells.

(ii) Given the weakness of the human dataset (see comments in Part II), elements of the results appear to be over-interpreted.

- In the Abstract, it is hard to justify the comment regarding ‘confirmed the relevance of this fungus-specific response’ given the dearth of information about it in the human setting (eg which cytokines are produced (most likely not IL-17), whether there is direct/indirect activation of γδ T cells). This is compounded by the fact that Malassezia is a skin fungus and yet these assays are performed on PBMC, so it’s not exactly a physiological setting – something I feel should be included as a caveat in the Discussion.

- In the Discussion, the current text states that ‘the relevance of this Malassezia-derived moiety as a specific agonist for γδ T cells is further underlined by the ability to activate human peripheral blood γδ T cells’. This is a major stretch, as just because there is an effect on human γδ T cells does not mean the mechanism or components are the same. It is therefore entirely unclear from the current limited dataset if the mechanism is conserved or not. Given the general lack of alignment between mouse and human γδ T cells, a similar mechanism would arguably be more surprising and certainly has not been proven. Of relevance, I don’t believe the involvement of a soluble factor from Malassezia has been shown in the human system – the manuscript only refers to use of heat-killed Malassezia in the PBMC assay. Furthermore, as stated above, a specific effect on γδ T cells (as opposed to an indirect effect, and/or an effect on both γδ T cells and other cells) has definitely also not been proven. So discussion of the human results as denoting a Malassezia-derived moiety that is a specific agonist for γδ T cells is premature.

(iii) A final query for the Authors is whether they can exclude the possibility that the ability of Malassezia-exposed Vγ4 T cells to sense a soluble Malassezia-derived component is TCR-dependent. Could it be that the initial stages of the Vγ4 response are cytokine driven (IL-23, IL-1) and TCR-independent, but the post-exposure, direct sensing of the soluble Malassezia component is influenced by changes in the TCR repertoire after initial exposure. A relevant experiment (whether practical or affordable) might be to assess the Vγ4 subset repertoire before and some time after 1st exposure to Malassezia. Major clonotypic focussing might imply antigen-specific selection and shaping of the response by exposure to the fungus. If this possibility can’t be excluded, this is something that could be highlighted in the Discussion.

(iii) A final point is that on occasion figure panels are called out in the text out of order, and I believe Figure 8F is not called out in the text at all.

Reviewer #2: Minor comment

1. Page 8 Ln 235 spelling, trangene positive

Reviewer #3: 1) The neutrophils numbers of the IL-17af-/- mice reach the level of WT mice ~day7. It’s interesting that at that point the fungal load in those mice then begins to drop, but they never clear the infection. It would be beneficial to discuss why this may be. Do the neutrophil levels drop again. Is IL-17 important for their retention and/or activity at this stage of infection?

2) gd Tcells are also known for being able to produce IFNgamma (as opposed to IL-17) and in some circumstances have been found to co-produce both IL-17 and IFNgamma in response to microbes. Because the IFNgamma producing gd Tcells are known to be more protective, it may be important to know whether IFNgamma may also be affected or even produced by the gd T cells in response to M.pach. It would be beneficial to add this to the discussion.

3) Your data shows that gd T cell deficient mice (Fig3) have a stronger initial recruitment of neutrophils and are able to eventually clear the M.pach infection compared to IL17af-/- mice (Fig1). This supports the involvement of a separate cell(s) in the Tcrd-/- mice that are responding to IL-17 to eventually clear M.pach skin infection. A discussion on this would be beneficial.

4) In Fig1C, the tissue inflammation pf the IL-17af-/- mice appears to go up visually from day 2 to day 4, whereas in the graph of the combined data (1B) it is shown to slightly drop or stay the same from day 2 to 4. And visually, there isn’t a big difference between day 4 and 7 in the images, but there is in the graph of the measured thickness. Are the images presented in 1C for the IL-17af-/- mice truly representative of the overall result depicted in Fig1B?

5) Human and mouse gd T cells are known to be different. Did you characterize which human gd Tcells were expanding in response to Malassezia? Did they induce IL-17? These experiments could add a lot of support towards a similar mechanism across hosts if included.

6) It is mentioned in the text that neutrophils are infiltrating into the skin as observed by histology (line 119). Please provide a higher magnification image and labeling to support this statement or remove the “and by histology” statement as there is already flow cytometry data demonstrating this.

7) Why was PBS used as a control for anti-Gr1 instead of an isotype antibody (Fig 7G-K)?

PLOS authors have the option to publish the peer review history of their article (what does this mean?). If published, this will include your full peer review and any attached files.

Reviewer #1: No

Reviewer #2: No

Reviewer #3: No

Figure Files:

Data Requirements:

Reproducibility:

References:

---

## [Editor Report · Decision Letter 1]

16 Dec 2023

Dear Dr. LeibundGut-Landmann,

We are pleased to inform you that your manuscript 'γδ T cells respond directly and selectively to the skin commensal yeast Malassezia for IL-17-dependent fungal control.' has been provisionally accepted for publication in PLOS Pathogens.

Best regards,

Amariliz Rivera

Academic Editor

PLOS Pathogens

Alex Andrianopoulos

Section Editor

PLOS Pathogens

Kasturi Haldar

Editor-in-Chief

PLOS Pathogens

orcid.org/0000-0001-5065-158X

Michael Malim

Editor-in-Chief

PLOS Pathogens

orcid.org/0000-0002-7699-2064

The authors have nicely addressed previous concerns. We appreciate the transparency in regards to the human data and agree that it is best to remove it from the paper. The murine studies are rigorous and convincing and can serve as foundation for further work in human cells once better methods and reagents are available.
---

## [Editor Report · Acceptance letter]

10 Jan 2024

Dear Dr. LeibundGut-Landmann,

We are delighted to inform you that your manuscript, " γδ T cells respond directly and selectively to the skin commensal yeast  * Malassezia *  for IL-17-dependent fungal control. ," has been formally accepted for publication in PLOS Pathogens.

Best regards,

Michael Malim

Editor-in-Chief

PLOS Pathogens

orcid.org/0000-0002-7699-2064